# Chemical shift prediction of RNA imino groups: application toward characterizing RNA excited states

Yanjiao Wang[1], Ge Han[1], Xiuying Jiang[1], Tairan Yuwen [2] & Yi Xue[1✉]

NH groups in proteins or nucleic acids are the most challenging target for chemical shift prediction. Here we show that the RNA base pair triplet motif dictates imino chemical shifts in its central base pair. A lookup table is established that links each type of base pair triplet to experimental chemical shifts of the central base pair, and can be used to predict imino chemical shifts of RNAs to remarkable accuracy. Strikingly, the semiempirical method can well interpret the variations of chemical shifts for different base pair triplets, and is even applicable to non-canonical motifs. This finding opens an avenue for predicting chemical shifts of more complicated RNA motifs. Furthermore, we combine the imino chemical shift prediction with NMR relaxation dispersion experiments targeting both $^{15}$N and $^{1}$H$^{N}$ of the imino group, and verify a previously characterized excited state of P5abc subdomain including an earlier speculated non-native G•G mismatch.

[1] School of Life Sciences, Tsinghua-Peking Joint Center for Life Sciences, Beijing Advanced Innovation Center for Structural Biology, Tsinghua University, Beijing, China. [2] Department of Pharmaceutical Analysis & State Key Laboratory of Natural and Biomimetic Drugs, School of Pharmaceutical Sciences, Peking University, Beijing, China. ✉email: yixue@mail.tsinghua.edu.cn

Chemical shift is the most valuable observable in NMR spectroscopy. It is easily accessible, can be precisely measured with high reproducibility, and most importantly is exquisitely sensitive to even a subtle change in biomolecular conformations. It has been well established that chemical shifts of proteins are strongly linked to their secondary structures, three-dimensional (3D) coordinates, and even dynamics[1–3]. To extract rich structural and dynamic information encoded in NMR chemical shifts, an accurate chemical shift predictor is the key. Methods for chemical shift prediction have been well developed for proteins over the past four decades based on ab initio quantum mechanical calculations[4,5], empirical data mining[6–8], or sequence homology[9]. For RNAs, such methodology development nevertheless lags markedly behind that for proteins. For a long time, empirical or semiempirical methods for RNA chemical shift prediction have focused on non-exchangeable protons[10–12]. Just in recent years, $^{13}$C chemical shift predictors with reasonable accuracy become available[13–15]. It has been demonstrated that chemical shifts of these two spins are sensitive to changes in the 3D conformation[16–18].

As a counterpart of protein amide group, RNA imino group serves as an excellent probe in NMR studies due to better dispersion property, less resonance broadening, and clearer connection with base pair types. The imino proton chemical shifts have long been serving as a sensitive indicator of the secondary structure. On the other hand, it remains a significant challenge to accurately predict chemical shifts of NH group in either proteins or RNAs mainly because this chemical group often participates in comprehensive intramolecular and intermolecular hydrogen bondings, as well as solvation effects, which are difficult to model due to their dynamic nature. For RNAs, such efforts are further hampered by insufficient imino resonance data and occasional assignment mistakes or nomenclature errors in the Biological Magnetic Resonance Bank (BMRB)[19]. As a result, there are currently no predictors for the imino group of RNAs, except for a tentative functional module in program LARMOR$^D$ (ref. [15]).

Here, we report a database-based imino chemical shift predictor for RNA A-form helical segments composed of only GC and AU Watson–Crick (WC) base pairs, as well as GU wobbles, in light of a premise that nucleic shielding of a base pair in helical context is predominantly determined by the central base pair and the two flanking base pairs immediately above and below it. The established lookup table can be used to accurately predict imino chemical shifts of RNA residues located in the center of base pair triplets. Ring-current (RC) contributions from aromatic rings of the nearest-neighboring base pairs can well reproduce imino chemical shifts in the lookup table, suggesting the semi-empirical method is promising in reliably predicting chemical shifts of more general RNA motifs. Using UUCG tetraloop as the structural model, we confirm the great potential of this method in predicting chemical shifts of noncanonical motifs. Finally, we demonstrate this chemical shift prediction approach can be of great help in the secondary structure determination of RNA excited states (ESs), when combined with $^{15}$N and $^1$H$^N$ NMR relaxation dispersion (RD) experiments.

## Results

### Imino chemical shift prediction of RNAs based on base pair triplets

Given that imino resonances stem from guanine and uridine only in base-paired regions, the base pair triplet within A-form helix (referred to as BP-triplet hereinafter, Fig. 1a) becomes the most common motif, in which imino resonances can be detected. It has been reported that the chemical shift of a non-exchangeable proton in RNA A-form helical regions can be predicted within an accuracy of root-mean-square deviation (r.m.

s.d.) 0.05 p.p.m. if it is located in the center of a WC BP-triplet[10]. Let us consider here the BP-triplet consisting of GC, AU, and GU base pairs. The total number of all possible BP-triplets capable of producing imino resonances is $6 \times 4 \times 6 = 144$. However, only 84 different BP-triplets with both $^1$H$^N$ and $^{15}$N imino chemical shifts can be extracted from BMRB. To address this data gap and avoid the impact of potentially erroneous data in BMRB, we prepared 30 unlabeled RNA hairpins, each containing a stretch of base pairs and an apical loop (Fig. 1b and Supplementary Table 1). These hairpin constructs were designed to cover all the 144 BP-triplet types and ensure at least two occurrences for each BP-triplet type.

Imino chemical shift data of these 30 RNA samples were collected under the same condition (10 mM sodium phosphate, 0.01 mM EDTA, pH 6.4, and 10 °C). These data were then combined with imino chemical shift data from BMRB database (in total 138 datasets updated to September 2018, Supplementary Table 1) to constitute the training dataset for chemical shift prediction. From the training dataset, we extracted all imino $^{15}$N and $^1$H$^N$ chemical shifts of guanine and uridine residues located in the center of BP-triplets. Chemical shift referencing errors were corrected by minimizing the overall chemical shift deviation of common motifs, including BP-triplets and UUCG apical loop (see "Methods" for details). With the exception of few outliers, imino resonances stemming from the same BP-triplet are clearly clustered within a narrowed region in a 2D spectrum, no matter how the surrounding sequence varies (Supplementary Fig. 1). These outliers were then trimmed off according to the three-sigma rule[13], and account for ~7% of the total data points (67 out of 920 for $^{15}$N, and 94 out of 1292 for $^1$H$^N$, see Fig. 1c). After reviewing these outliers, we found that they can be attributed to multiple factors, such as distorted conformations, long-range interactions, misassignments, and unusual buffer conditions (such as pH, ionic strength, temperature, and interaction with divalent metal ions). In the end, we established a lookup table that relates each BP-triplet type to the average experimental imino chemical shifts of multiple occurrences of that specific BP-triplet (Supplementary Table 2). This table can be used to predict imino chemical shifts in helical regions of RNAs. For the training dataset, such prediction yields a high accuracy after outliers are removed: r.m.s.d.($^{15}$N) = 0.169 p.p.m., r.m.s.d.($^1$H$^N$) = 0.073 p.p.m. (Fig. 1c), which is unsurprisingly much better than the result of LARMOR$^D$ (Supplementary Fig. 2).

To test the performance of the predictor, we separately compiled a testing dataset comprising the latest BMRB data (seven entries) and experimental data from ten additional unlabeled hairpin samples measured in this work and four labeled samples measured in prior works (Supplementary Table 3). The prediction using the testing dataset still leads to a high accuracy: r.m.s.d.($^{15}$N) = 0.193 p.p.m., r.m.s.d.($^1$H$^N$) = 0.097 p.p.m. (Fig. 1d). It is worth noting that the prediction accuracy does not depend on the definition of the training dataset. Indeed, a similar result is achieved when different strategies are used to split the training and testing datasets, for instance, using only BMRB data or using only data from 30 hairpin samples as the training dataset (Supplementary Table 4). Unlike the data we collected, BMRB data were acquired under different conditions (temperature, pH, and salt concentration). Our result indicates that the temperature and buffer condition have little influence on our chemical shift prediction. It is very likely that the re-referencing procedure "absorbed" the chemical shift perturbation caused by varied conditions. To confirm it, we measured $^1$H–$^{15}$N 2D spectra at 10 and 25 °C for nine hairpin samples. All imino resonances show a sizable and largely uniform upfield shift on the $^1$H dimension (Supplementary Fig. 3). After re-referencing, the chemical shifts at the two temperatures agree

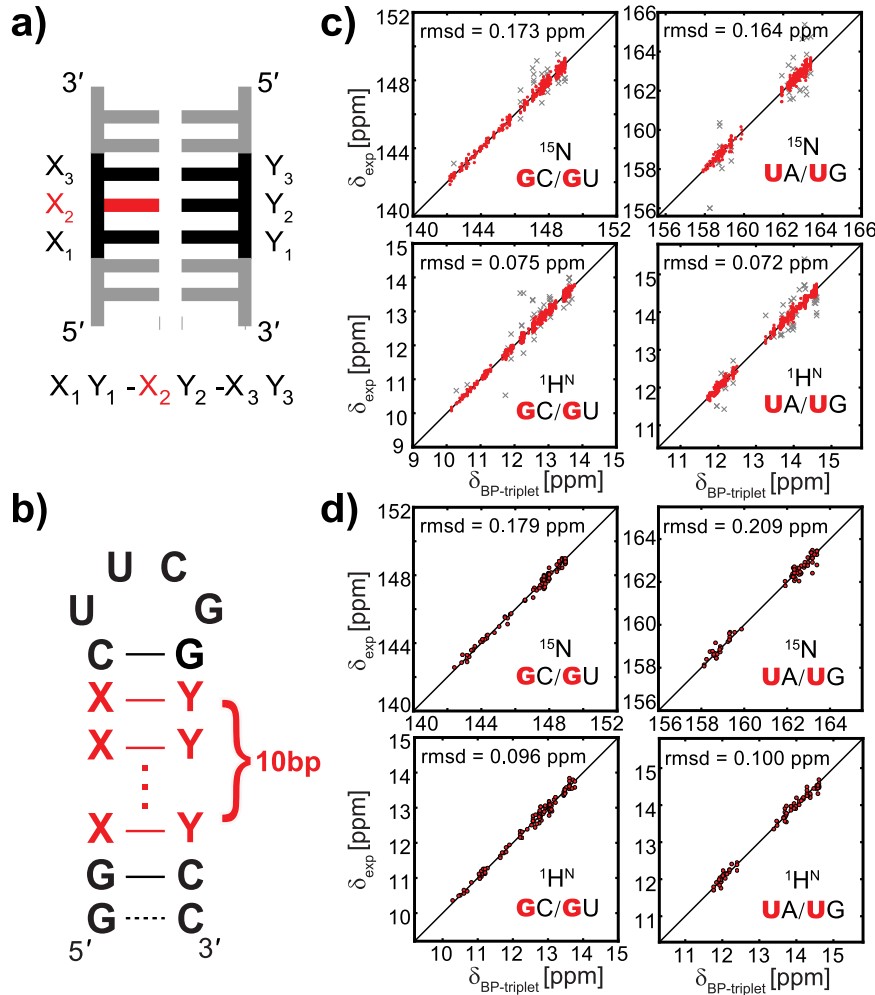

**Fig. 1 Prediction of imino chemical shifts using the base pair triplet (BP-triplet). a** Definition of BP-triplet where the nucleotide under measurement is colored in red. Each BP-triplet can be described by a three-field code: the left field represents 5′ neighboring base pair judging from the nucleotide of interest, while the right field represents the base pair to the 3′ direction. **b** RNA hairpins used for collecting imino chemical shift data. Base pairs in the stem region (red) are designed to produce as many different BP-triplets as possible. For a few hairpin samples, the UUCG apical loop is replaced with an AACGU pentaloop, and eight base pairs instead of ten are placed in the stem region (see Supplementary Tables 1 and 3). **c** Correlation between predicted and experimental $^{15}$N (upper panel) and $^1$H$^N$ (lower panel) chemical shifts for BP-triplets centered with GC/GU (left) and UA/UG (right) in the training dataset. The outliers (x marker in gray) are identified by the $3\sigma$ rule and excluded from statistics. The overall rms deviations for $^{15}$N and $^1$H$^N$ are 0.169 and 0.073 p.p.m., respectively. **d** Correlation between predicted and experimental $^{15}$N (upper panel) and $^1$H$^N$ (lower panel) chemical shifts for BP-triplets centered with GC/GU (left) and UA/UG (right) in the testing dataset. The overall rms deviations for $^{15}$N and $^1$H$^N$ are 0.193 and 0.097 p.p.m., respectively.

with each other very nicely. The r.m.s.d. values for $^{15}$N and $^1$H$^N$ are 0.079 and 0.029 p.p.m., respectively, well below the uncertainty of our predictor.

Since A-form helix is one of the most stable structural motifs in biomacromolecules, these BP-triplet chemical shift data provide us with an excellent opportunity to look into the relationship between RNA structure and chemical shift. For convenience, the BP-triplet lookup table can be visualized as an imino chemical shift map (Fig. 2 and Supplementary Fig. 4). As shown in this map, imino resonances of guanines from GC or GU show larger dispersion while those of uridines from UG are dispersed the least. Besides, GC and GU resonances are largely distributed along a straight line with a slope of 2, whereas such a pattern is not seen in UA and UG clusters. Can these features be interpreted by any computational models of chemical shift?

**The ring-current effect is the dominant factor of imino chemical shifts.** It has been demonstrated that chemical shift of the non-exchangeable proton in nucleic acids can be quantitatively predicted to a good approximation by the semiempirical model[11], in which the total chemical shift of a proton is the sum of the intrinsic shift $\delta_{\text{intrin}}$ that reflects the intrinsic shielding effect of the local electronic structure, shifts from the RC effect of all nearby aromatic rings $\sum\delta_{\text{rc}}$, shifts from the local magnetic anisotropy effect $\sum\delta_{\text{ma}}$, and shifts from the electric-field-induced (EF) polarization $\sum\delta_{\text{ef}}$. The magnetic anisotropy term $\sum\delta_{\text{ma}}$ can be absorbed into the RC contribution[20]. Therefore, the chemical shift of a nucleus in the central base pair of a BP-triplet becomes $\delta_{\text{calc}} = \delta_{\text{intrin}} + \sum\delta_{\text{rc}} + \sum\delta_{\text{ef}}$, where $\delta_{\text{intrin}}$ represents the intrinsic chemical shift of this central base pair, $\sum\delta_{\text{rc}}$ and $\sum\delta_{\text{ef}}$ are the RC contribution and the EF-induced contribution, respectively, from 5′- and 3′-nearest-neighboring base pairs. The electrostatic contribution $\sum\delta_{\text{ef}}$ was found to be negligible for non-exchangeable protons in RNA structures[11], and we have confirmed that this conclusion is applicable to imino protons in BP-triplets as well (see below). The parameter set of RC and EF for proton was initially proposed by

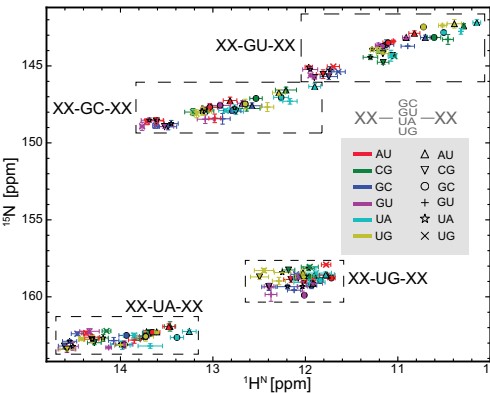

**Fig. 2 Imino chemical shift map of BP-triplets.** The average imino resonance of each BP-triplet is shown as a marker on the map. Each BP-triplet can be described as a triplet code (see Fig. 1a): the dashed boxes depict the chemical shift distribution range of the central guanine or uridine residue as defined by the first letter of the middle code; the 5′ and 3′ neighboring base pairs are defined by color and shape of makers, respectively. The error bars indicate one standard deviation (s.d.; the corresponding sample sizes are provided in Supplementary Table 2).

**Table 1 Ring-current intensity factors used to calculate RC shifts.**

| Ring | Gua-5 | Gua-6 | Ade-5 | Ade-6 | Cyt | Ura |
|---|---|---|---|---|---|---|
| Previous intensity factor (N and H) | 0.81 | 0.49 | 0.95 | 0.83 | 0.31 | 0.24 |
| Calibrated intensity factor (N) | 2.60 | 0.11 | 3.57 | 0.06 | 0.84 | 1.32 |

Giessner-Prettre and coworkers[21] (termed GP set), and later re-parameterized by Case group[20] (DC set) and recently by Vendruscolo group[22] (MV set). For $^{15}$N and $^{13}$C, however, there is no reliable RC parameter set at present, and even no reasonable EF calculation model. The semiempirical model has been used to interpret experimental non-exchangeable proton chemical shifts of nucleic acids since 1970s (refs. [23–26]). However, similar works on the imino proton were rarely reported and limited to very few experimental data[27,28].

Here, we constructed 3D models of A-form RNA using RNAComposer[29], and calculated RC shifts caused by aromatic rings of the two nearest-neighboring base pairs, using DC parameter set for both $^{1}$H$^{N}$ and $^{15}$N (Table 1, see "Methods" for details). Strikingly, a good correlation was observed between the calculated RC shifts and the BP-triplet lookup table (Fig. 3a), indicating that RC contribution is the dominating factor for chemical shift variations caused by different flanking base pairs. This conclusion can be further confirmed by comparing EF shift and RC shift for $^{1}$H$^{N}$ of each BP-triplet (Supplementary Fig. 5). Indeed, neither expanding BP-triplet to five consecutive base pairs (Supplementary Table 5) nor including EF contributions (Supplementary Fig. 6) can improve correlation to a meaningful extent. Of note, $^{15}$N spin of uridine shows poor correlation, and $^{15}$N spin of guanine in GU base pair shows a slope clearly deviated from 1.0 (Fig. 3a), which could be attributed to several factors, such as the dynamics of GU base pair, impropriate base plane geometry, and not fully optimized RC parameters (particularly for $^{15}$N).

To assess the influence of base plane geometry, we calculated RC shifts using BP-triplet fragments extracted from RNA crystal structures in Protein Data Bank (PDB) with resolution better

than 2 Å, as well as from A-form helix models built by 3DNA that can model only WC base pairs. The calculated RC shifts from crystal structures show high diversity for each specific BP-triplet (Supplementary Fig. 7a), indicating the geometry parameters of each BP-triplet in crystal structures are far from uniform. After taking the average of the RC shifts from the same BP-triplet, the crystal structures lead to a comparable agreement on $^{15}$N, but moderately worse agreement on $^{1}$H$^{N}$ with BP-triplet lookup table, as compared to RNAComposer structures (Supplementary Fig. 7b). Interestingly, the result of the 3DNA model shows slightly better agreement (Supplementary Fig. 7c, d). Further, we examined rigid-body parameters of base pairs and base pair steps of these BP-triplet models (Supplementary Fig. 8). The 3DNA structures show higher similarity with crystal structures in terms of base pair geometry. All these results suggest that RC calculation could be helpful in the structure refinement of nucleic acids. It is worth mentioning that, although the rigid-body geometries of the RNAComposer model markedly deviate from those of crystal structures (especially for BP-triplets involving GU wobble), the resulting RC shifts are not remarkably different from that of crystal structures (Supplementary Fig. 9).

The RC parameter set used above was parameterized against proton[20], and may not be applicable to $^{15}$N. Since the RC contributions are dominant, we re-parameterized RC parameters for $^{1}$H$^{N}$ and $^{15}$N to maximize the agreement with data in the lookup table (see "Methods"). As expected, parameters of $^{1}$H$^{N}$ show only minor changes after optimization and lead to slightly improved correlation. In contrast, parameters of $^{15}$N deviate from the original values significantly (Table 1), and the agreement between the semiempirical result and the lookup table becomes noticeably better (Fig. 3b). In the following RC calculations, the original DC parameters for proton and the calibrated DC parameters for nitrogen will be used.

Remarkably, the imino chemical shift map generated by the RC calculation (Supplementary Fig. 10) encapsulates the conspicuous features observed in the experimental map (Fig. 2), including the dispersion range and the cluster slope. These results provide an important foundation for predicting NH and even CH chemical shifts of more complicated structural motifs, such as those involving noncanonical base pairs, bulges, and loops.

**Experimental imino chemical shifts of BP-triplets are decomposable.** The semiempirical calculation described above provides a practical way to decompose chemical shifts in the BP-triplet lookup table, as the RC contributions from 5′ base pair and 3′ base pair can be calculated separately (see "Methods"). In doing so, each imino chemical shift in the lookup table can be split into three components as shown in Table 2: (1) the intrinsic chemical shift of the central base pair; (2) the contribution to the specific central base pair from the 5′ base pair; (3) the contribution to the specific central base pair from the 3′ base pair. Consequently, the intrinsic chemical shifts of each base pair (GC, UA, GU, and UG) can be determined in a straightforward manner, providing corrections to the previously published results that can deviate from the current values by up to 0.3 p.p.m. (Table 2, numbers in brackets).

To verify the effectiveness of the decomposed shifts, we reconstructed the imino chemical shifts of all 144 BP-triplets using these values. The reconstructed chemical shifts are in excellent agreement with the lookup table, and r.m.s.d. values for $^{15}$N and $^{1}$H$^{N}$ are 0.131 and 0.059 p.p.m., respectively. More importantly, the reconstructed BP-triplet chemical shifts can predict experimental imino data very well, with r.m.s.d. only marginally increased (Fig. 4).

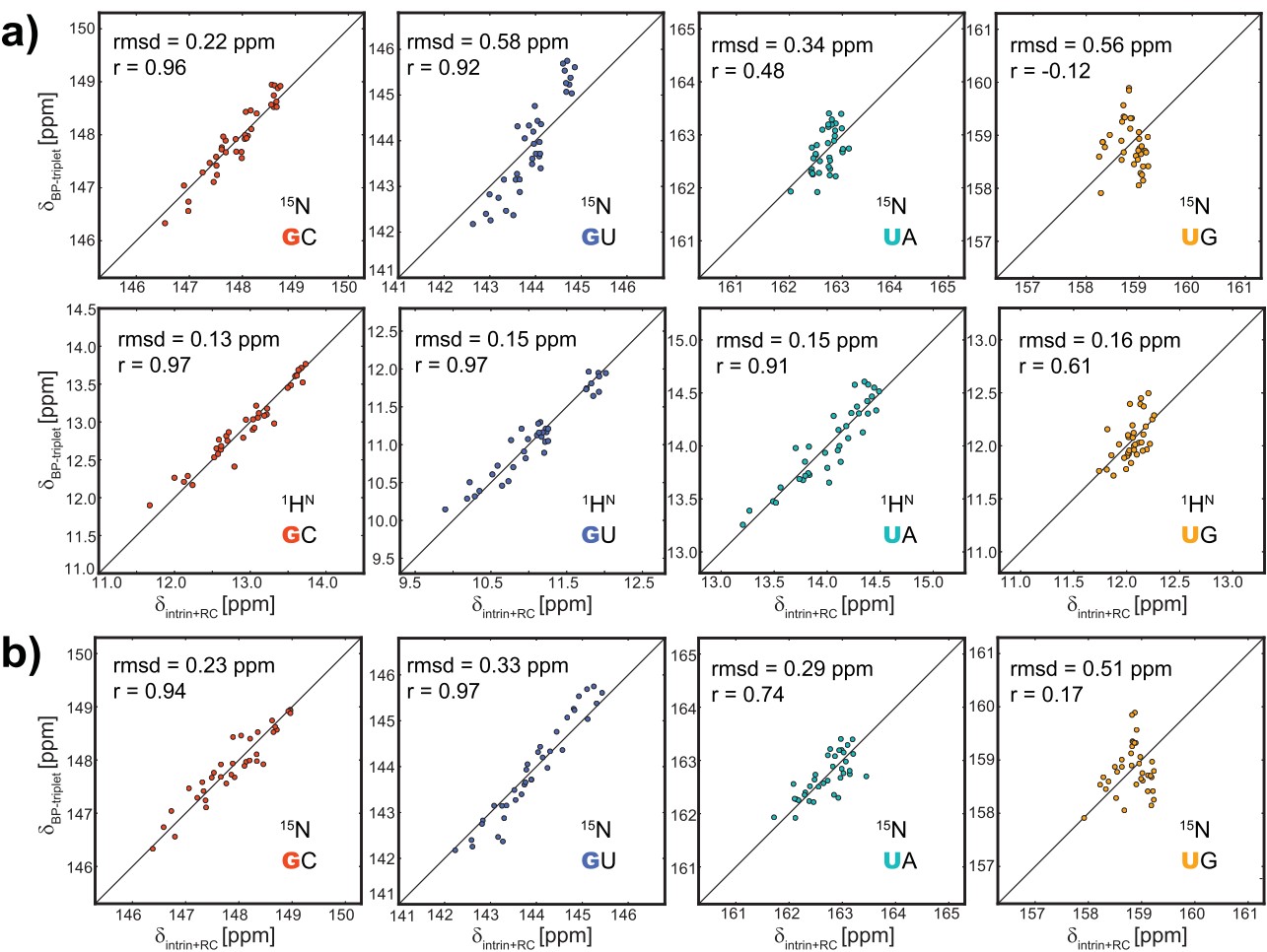

**Fig. 3 Correlation between chemical shifts calculated by the semiempirical model and chemical shifts in the BP-triplet lookup table.** The structure models were generated by RNAComposer. Chemical shift data in the BP-triplet table are divided into four groups (GC: red; GU: blue; UA: cyan; UG: orange) according to types of the central base pair. **a** Correlation between chemical shifts calculated by the semiempirical model using original RC parameters (DC set) and chemical shifts in the BP-triplet lookup table. The upper four subplots show the correlation for $^{15}$N, and the lower four show the correlation for $^1$H$^N$. **b** Correlation between $^{15}$N chemical shifts calculated by the semiempirical model using calibrated RC parameters and $^{15}$N chemical shifts in the BP-triplet lookup table. The semiempirical calculation was carried out by summing up the intrinsic chemical shift of the central base pair and RC contributions from two neighboring base pairs. For each subplot, the intrinsic chemical shift was adjusted so that the semiempirical result and the chemical shift data from the BP-triplet table give the same mean value.

**Table 2 Decomposed chemical shift contributions from central base pairs and 5′/3′ neighboring base pairs.**

| Intrinsic | GC [p.p.m.] | | UA [p.p.m.] | | GU [p.p.m.] | | UG [p.p.m.] | |
|---|---|---|---|---|---|---|---|---|
| | **N** | **H** | **N** | **H** | **N** | **H** | **N** | **H** |
| | 149.92 ± 0.11 | 14.01 ± 0.06 (13.7)[a] | 164.03 ± 0.12 | 14.85 ± 0.06 (14.8)[a] | 146.15 ± 0.13 | 12.20 ± 0.05 (12.5 ± 0.1)[b] | 159.69 ± 0.16 | 12.27 ± 0.07 (12.2 ± 0.1)[b] |
| 5′ GC | −0.25 | −0.36 | −0.37 | −0.24 | −0.32 | −0.38 | −0.30 | −0.10 |
| 5′ UA | −1.29 | −1.04 | −0.25 | −0.78 | −1.38 | −0.97 | −0.73 | −0.29 |
| 5′ GU | −0.37 | −0.22 | −0.08 | −0.26 | −0.27 | −0.18 | −0.10 | −0.07 |
| 5′ UG | −1.02 | −0.71 | −0.24 | −0.36 | −1.59 | −0.81 | −0.88 | 0.05 |
| 5′ AU | −0.49 | −0.20 | −0.59 | −0.47 | −0.55 | −0.28 | −0.79 | −0.32 |
| 5′ CG | −1.16 | −0.72 | −0.58 | −0.54 | −1.06 | −0.88 | −1.03 | −0.20 |
| 3′ GC | −1.69 | −0.77 | −1.10 | −0.69 | −2.08 | −0.75 | −0.01 | −0.20 |
| 3′ UA | −0.97 | −0.19 | −0.81 | −0.10 | −0.63 | −0.05 | −0.24 | −0.04 |
| 3′ GU | −1.15 | −0.77 | −0.69 | −0.49 | −2.09 | −0.88 | 0.13 | 0.01 |
| 3′ UG | −0.75 | −0.07 | −1.27 | −0.14 | −0.51 | −0.18 | −0.73 | −0.07 |
| 3′ AU | −2.18 | −1.05 | −1.48 | −0.87 | −2.62 | −1.07 | −0.36 | −0.23 |
| 3′ CG | −0.77 | −0.21 | −0.53 | −0.02 | −0.25 | −0.14 | −0.18 | 0.17 |

The table header shows the central base pair types and the relevant spins. The intrinsic chemical shifts reported by previous studies ([a], ref. [27] and [b], ref. [28]) are shown in brackets.

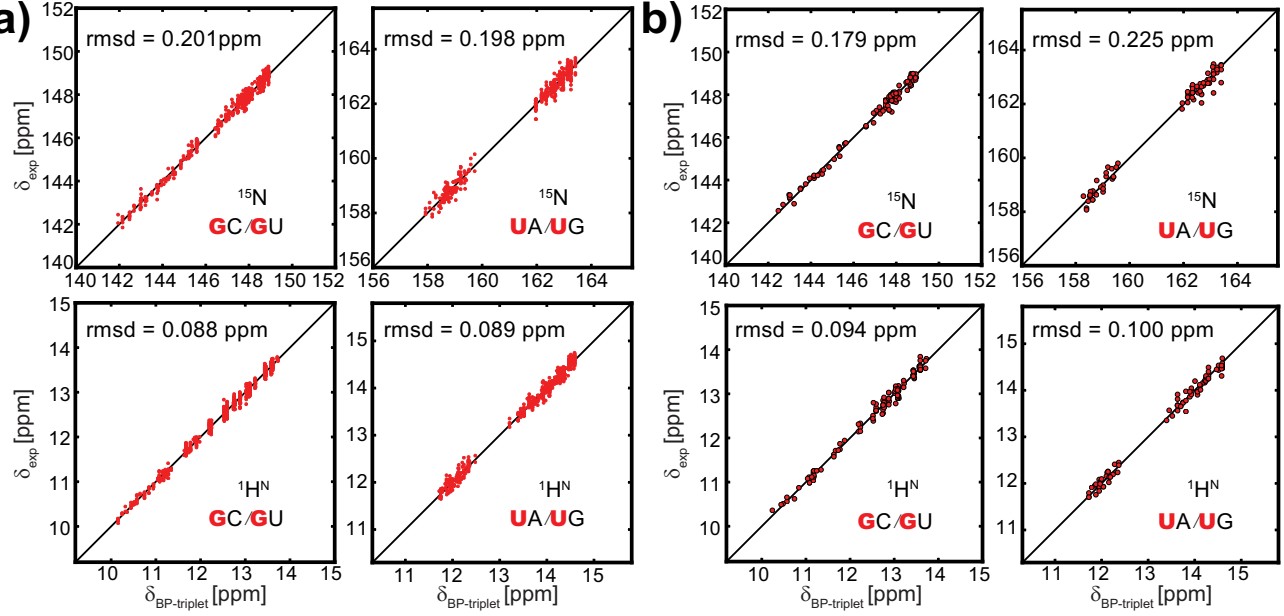

**Fig. 4 The reconstructed imino chemical shifts using the decomposed shifts. a** Correlation between reconstructed and experimental $^{15}N$ (upper panel) and $^{1}H^{N}$ (lower panel) chemical shifts for BP-triplets centered with GC/GU (left) and UA/UG (right) in the training dataset. **b** Correlation between reconstructed and experimental $^{15}N$ (upper panel) and $^{1}H^{N}$ (lower panel) chemical shifts for BP-triplets centered with GC/GU (left) and UA/UG (right) in the testing dataset.

The decomposed imino chemical shifts can facilitate the semiempirical calculation. For instance, when one of the neighboring base pairs is a noncanonical one, we only need to perform the semiempirical calculation on this noncanonical motif, and add up the result with the decomposed chemical shifts of the other neighboring base pair and the central base pair. This approach is preferred, as part of semiempirical result is replaced with the decomposed value that is presumably more accurate. In some sense, it can be viewed as a hybrid method combining both the semiempirical approach and the lookup table. We will demonstrate this method below.

**Semiempirical method is applicable to noncanonical motifs.** We chose the UUCG tetraloop as the noncanonical motif to test the semiempirical method. UUCG tetraloop is one of the most stable structural motifs in RNAs and thus the structure models with high fidelity are available. Besides, we have collected many experimental data for 5′-CUUCGG-3′ motif with different base pairs appended to the end (Fig. 5). Three NMR structures (PDB code: 2KOC, 2M4Q, and 5IEM) and two high-resolution X-ray structures (PDB code: 1F7Y and 5Y85) were used in the semi-empirical calculations (Supplementary Table 6). For the UUCG motif, the imino chemical shifts of the guanine in the central CG base pair are the sum of three components: (1) RC shift of UUCG tetraloop (assuming EF shift is ignorable); (2) the intrinsic che-mical shift of the central CG base pair; (3) the contribution from the 3′-neighboring base pair. The last two items can be found in Table 2. Impressively, the calculated imino chemical shifts of the central guanine based on 2KOC structure, a state-of-the-art NMR structure of UUCG motif that incorporates all currently accessible NMR experimental restraints[30], are in excellent agreement with the experimental data (Fig. 5). In contrast, the other two NMR structures (2M4Q and 5IEM) result in a much worse correlation with the experimental result. This is not surprising because these two structures were solved using considerably fewer restraints, and also do not specifically target UUCG motif. For the two

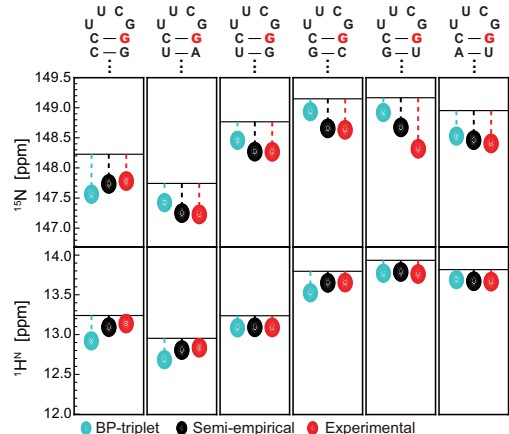

**Fig. 5 Comparison of $^{15}N$ (upper panel) and $^{1}H^{N}$ (lower panel) chemical shifts for the guanine in the CG closing base pair of UUCG tetraloops.** The secondary structures of UUCG motifs are shown on top of each panel, and the guanine of interest is highlighted in red. The horizontal line in each subplot marks the sum of chemical shift contributions from the central CG base pair and the base pair immediately below it. Resonances colored in cyan and black represent the total chemical shifts of the highlighted guanine by including the contribution from UUCG tetraloop through two approaches: prediction using the BP-triplet table (cyan); RC shift calculation using the calibrated parameter set (black). The resonances colored in red represent experimental values.

crystal structures, good correlations are also achieved between calculated chemical shifts and experimental ones.

To examine whether EF contribution can be safely ignored, we calculated $^{1}H^{N}$ EF shifts as described above. Indeed, for 2KOC and the two crystal structures, the calculated chemical shifts show only small changes as compared with the case when EF is absent (Supplementary Table 6). Although EF calculation of $^{15}N$ is not feasible, it is likely ignorable as well, given that $^{15}N$ and $^{1}H^{N}$ are

adjacent in space. In addition, we recalculated $^{15}N$ chemical shift of the guanine using the uncalibrated RC parameters. For 2KOC and the two crystal structures, the r.m.s.d. values become considerably elevated (Supplementary Fig. 11 and Supplementary Table 6), providing additional validation for our calibrated RC parameters of $^{15}N$.

**Imino chemical shift prediction helps to determine RNA excited states.** Predicting imino chemical shift from RNA secondary structure has multiple applications, such as facilitating (or validating) imino resonance assignment or RNA secondary structure determination. Here, we demonstrate an application involving secondary structure determination of RNA "ESs" that form through reshuffling base pairs in and around noncanonical motifs. RNA ESs involving the local rearrangement of the secondary structure are of great interest[31,32] as they are linked to functional regulation[33,34], enzymatic catalysis[34], ligand binding[35–37], and folding/unfolding[38,39]. These reshuffling motions usually fall into microsecond to millisecond time regime since only a few base pairings are changed during the exchange process. NMR RD approach has been proved to be very powerful in characterizing these low-abundance and short-lived ESs on per-residue basis[40]. Prior work established the utility of $^{15}N$ NMR RD to characterize ESs in large and complex RNAs[38,41,42]. However, there remains significant ambiguity in interpreting imino $^{15}N$ chemical shifts. Here, we extend this approach to include $^{1}H^{N}$ RD measurement and also take advantage of our chemical shift prediction approach to characterize ESs to a much greater degree of certainty.

In proteins, RD measurement of $^{1}H^{N}$ has been carried out for decades using CPMG or $R_{1\rho}$ experiments with the aid of sample deuteration[43–45]. Very recently, two CEST-based RD experiments have been developed to measure protein amide proton without sample deuteration[46,47]. These experiments can be directly applied to uniformly isotope-labeled RNAs, extending $^{1}H^{N}$ RD measurement from small and unlabeled RNAs[48] to larger RNAs. Proton per se is a very attractive probe for RD measurement, allowing detection of faster conformational exchange and lower population species due to higher applicable spin-lock power and wider dispersion range of proton chemical shift. In comparison with non-exchangeable protons, imino protons are particularly attractive for RNAs because their chemical shifts span ~5 p.p.m. and exhibit a characteristic distribution range for each base pair type (Fig. 2). Using the BP-triplet lookup table, one can predict $^{15}N$ and $^{1}H^{N}$ chemical shift changes of the central guanine or uridine due to all possible single-nucleotide register shifts in each BP-triplet (Fig. 6 and Supplementary Table 7). When a guanine or uridine changes its base pair type in response to the secondary structure switching between the ground state (GS) and the ES, $^{1}H^{N}$ shows chemical shift change roughly four times larger than $^{15}N$, which in turn is translated into higher RD signal. Even for switching without the change of base pair type, $^{1}H^{N}$ has more chances to experience pronounced chemical shift difference. Further, when both $^{15}N$ RD and $^{1}H^{N}$ RD are measured, the secondary structure information of ES can be derived from comparing the imino resonance location of ES in the chemical shift map with those predicted by presumed ES secondary structures (Fig. 2). We applied this strategy to a previously characterized ES of P5abc[38], a subdomain of the Tetrahymena group I intron ribozyme.

**Characterizing excited states of P5abc RNA.** Our prior work showed that in the absence of $Mg^{2+}$ P5abc undergoes secondary structure reshuffling in millisecond time scale between a dominant unfolded form and a ~3% populated folding intermediate,

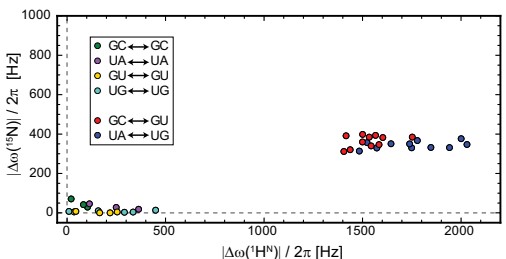

**Fig. 6 Predicted chemical shift change (absolute value in Hz, assuming 800 MHz spectrometer) caused by all possible one-nucleotide sliding in register of BP-triplets (see Supplementary Table 7 for the full list).** The data points are clearly clustered into two regions: the lower left region represents BP-triplet sliding during which the central base pair types remain the same; the middle right region represents BP-triplet sliding during which the central base pair types switch between GC and GU or between UA and UG.

through a single-nucleotide shift in register within P5c stem (Fig. 7a). In this ES, a noncanonical G•G mismatch is likely formed judging from $^{15}N$ RD of these two guanines. However, $^{15}N$ RD data alone cannot rule out other possibilities, such as the unpaired bases, and thus the use of $^{1}H^{N}$ RD is highly desirable.

We first repeated $^{15}N$ $R_{1\rho}$ measurement and also conducted $^{15}N$ CEST experiment. After fitting data globally to a simple two-state model, the resulting $\Delta\omega$ ($=\omega_{ES} - \omega_{GS}$) values from the two experiments are very close to each other (Fig. 7b, and Supplementary Figs. 12 and 13), and are also in excellent agreement with the previous result of $^{15}N$ $R_{1\rho}$ (ref. 38). Next, we carried out the TROSY-based imino $^{1}H^{N}$-CEST experiment with longitudinal relaxation optimized[49]. This experiment separates $^{1}H^{N}$ signal into $^{1}H^{N}(N^{\alpha})$-component and $^{1}H^{N}(N^{\beta})$-component, and the difference CEST profile is produced by subtracting the profile of one component from the other to completely suppress the undesired NOE dips[47,49]. Indeed, we observed minor dips in $^{1}H^{N}$ CEST profiles of four residues, as well as a single asymmetric dip from G175 due to small $\Delta\omega$ (Fig. 7c). The $\Delta\omega$ values were obtained by globally fitting to the two-state model. The imino $^{15}N$ and $^{1}H^{N}$ chemical shifts in the invisible ES were thus obtained by summing up $\Delta\omega$ and corresponding chemical shifts in GS (Supplementary Table 8).

With the newly acquired $^{1}H^{N}$ RD data, we can verify the previously found ES of P5abc, and resolve the ambiguity of G•G mismatch. The location of an ES imino resonance in the 2D spectrum immediately tells us the central base pair type. With a reliable imino chemical shift predictor based on BP-triplet, the information of triplet base pairs rather than just the central base pair can be derived, providing strong restraints for secondary structure determination of ES. Among five residues with RD signals, $^{15}N$ and $^{1}H^{N}$ chemical shifts of G174$^{ES}$ can be immediately predicted because this residue is located in a BP-triplet from the lookup table. Indeed, the experimental imino chemical shifts of G174$^{ES}$ are in excellent agreement with the predicted values (Fig. 7d and Supplementary Table 8). The other four residues in ES, nevertheless, are located in BP-triplets that do not exist in the table as they involve open bases or non-GU mismatches. Of course, significant efforts are required in the future to extend our prediction tool to noncanonical BP-triplets. At present, we resorted to a workaround instead. Specifically, we prepared two additional hairpin samples to produce the desired noncanonical BP-triplets: a GG1 hairpin with UUCG tetraloop (Supplementary Fig. 14a) for G164$^{ES}$, G176$^{ES}$, and G175$^{ES}$, where a G•G mismatch is included, and a hairpin with pentaloop (Supplementary Fig. 15) for U167$^{ES}$, where an A•U mismatch is

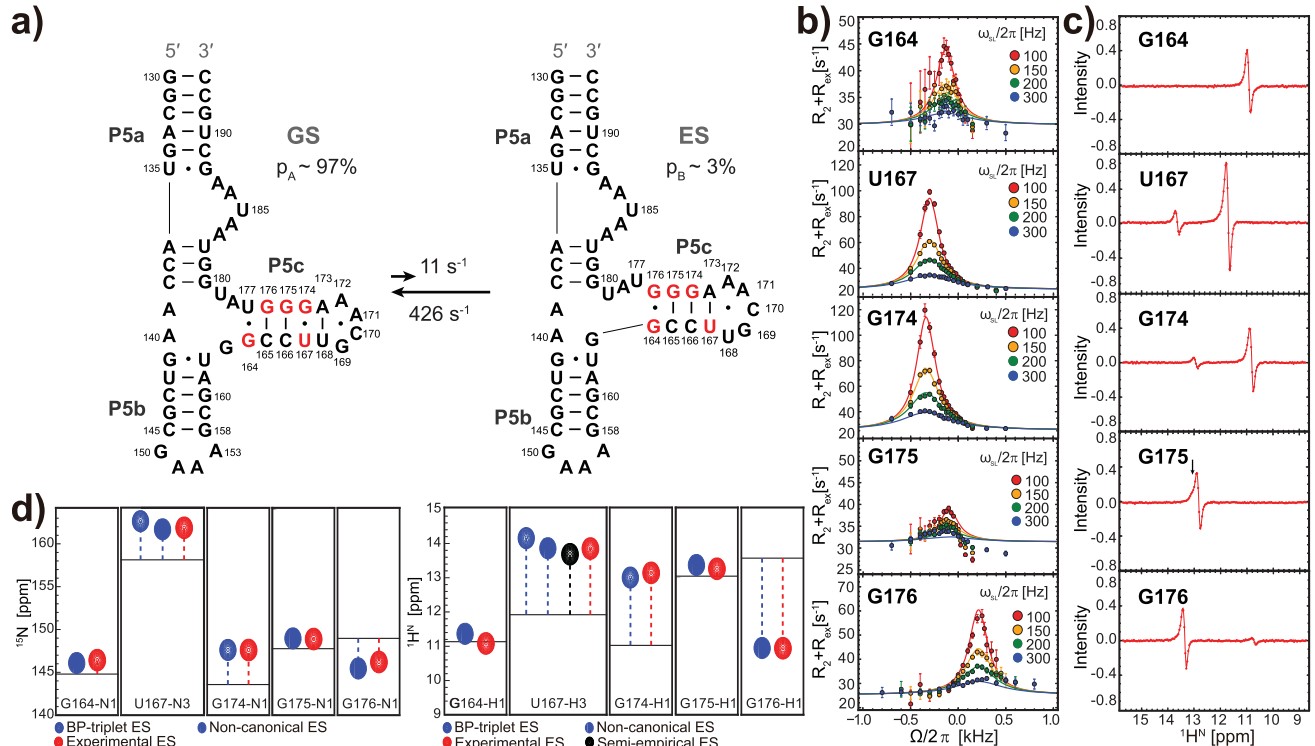

**Fig. 7 Verification of P5abc^ES secondary structure by using ¹⁵N RD and ¹H^N CEST, with the help of imino chemical shift prediction. a** Secondary structure reshuffling of P5abc in the absence of Mg²⁺. Residues showing pronounced RD are colored in red. G164•G176 is a speculated non-native mismatch in the previous study. **b** Off-resonance ¹⁵N $R_{1\rho}$ profiles of relevant residues in P5c stem. The on-resonance ¹⁵N $R_{1\rho}$ profiles and ¹⁵N CEST profiles are available in Supplementary Figs. 12 and 13, respectively. The error bars represent standard deviations (s.d.) estimated using Monte Carlo simulation with 50 iterations. **c** ¹H^N CEST difference profiles of the same set of residues in P5c stem. The profile of G175 shows an asymmetrical dip caused by a small $\Delta\omega$ (black arrow). **d** Depiction of imino ¹⁵N (left panel) and ¹H^N (right panel) chemical shifts of the excited state relative to the ground state (horizontal lines), as measured by RD (red resonances), and predicted by BP-triplet table (blue resonances), individual noncanonical BP-triplet (solid blue resonances), or semiempirical method (black resonance).

included. Putting all results together, the predicted imino chemical shifts show excellent agreement with the experimental values derived from RD experiments (Fig. 7d and Supplementary Table 8), with r.m.s.d. 0.24 p.p.m. for ¹⁵N and 0.13 p.p.m. for ¹H^N. Remarkably, the non-native G•G mismatch speculated in the previous study[38] is now secured, as both ¹⁵N and ¹H^N chemical shifts of G•G mismatch in ES are in line with the predicted results. Strictly speaking, BP-triplets of G164^ES and G176^ES are not exactly the same as those produced by GG1 hairpin, as G and U adjacent to G•G mismatch form GU wobble in GG1 hairpin (Supplementary Fig. 14), whereas this GU wobble is unlikely formed[38] in the ES (Fig. 7a).

It is intriguing to apply semiempirical calculation to non-canonical BP-triplets of P5abc, but this is hampered by the lack of high-resolution structure of P5abc^ES. Here, we turn to the crystal structure of folded P4–P6 (PDB code: 1GID), using the P5c loop region for the semiempirical calculation regardless of the involvement of Mg²⁺ and tertiary contacts. Unlike the other cases we handled above, the EF shift of U167-H3 contributed from P5c pentaloop is as large as 0.16 p.p.m., and including EF effect considerably improves the agreement with the experimental result (Fig. 7d). The U167-N3 chemical shift calculated by only RC differs from the experimental value by ~1.0 p.p.m., likely due to the lack of ¹⁵N EF contribution whose calculation is not feasible at present.

## Discussion
We have established a BP-triplet lookup table that makes a connection between imino chemical shifts of a base pair and the

BP-triplet where the base pair resides in the center. This table can be used to accurately predict ¹H^N and ¹⁵N imino chemical shifts for RNA helical segments composed of only WC base pairs and GU wobbles. The semiempirical analysis indicates RC contributions from two nearest-neighboring base pairs are responsible for chemical shift variations of BP-triplets, suggesting the semi-empirical model is a promising method for predicting chemical shifts of noncanonical motifs. The effectiveness of this method was then proven by using UUCG motif. In the end, we performed joint measurement of ¹⁵N RD and ¹H^N CEST, and successfully verified the secondary structure of P5abc^ES by virtue of imino chemical shift prediction. Particularly, a previously speculated non-native G•G mismatch is confirmed, which helps stabilize P5abc^ES as a folding intermediate[38]. Relatedly, the non-native interactions have been observed in folding intermediates of protein[50,51].

The imino chemical shift prediction based on our BP-triplet lookup table is only applicable to the A-form region made of WC base pairs and GU wobbles. For noncanonical motifs, the central base pair is typically adjacent to noncanonical base pairs, bulges, loops, and junctions. The semiempirical method is proven to be particularly helpful in this scenario. We found that both the RC and EF shifts are sensitive to minor conformational changes of an RNA, and thus an accurate structural description of noncanonical motif in static form or (more often) ensemble form is required for reliable semiempirical calculations. Conversely, chemical shifts can serve as highly effective structural restraints with the aid of semiempirical method. To this end, accurate RC and EF models for ¹⁵N and ¹³C are in urgent need.

Compared with joint RD measurement of [15]N and [1]H[N], the combination of [13]C RD and [1]H RD has advantages in measuring unpaired residues, and has been recently employed to characterize RNA ESs[35,52]. These prior studies, nevertheless, require spin-selective labeling and are also hampered by relatively narrow range of proton resonances and less clean-cut relationship between experimental CH chemical shifts and structural information. In contrast, the combination of [15]N RD and [1]H[N] RD is applicable to uniformly labeled RNAs. More importantly, with the aid of imino chemical shift prediction, the joint [15]N/[1]H[N] RD measurement provides valuable secondary structure information for invisible RNA ESs. A critical step for ES verification is to design often more than one constructs to trap ES, a strategy named mutate and chemical shift fingerprint (MCSF)[53]. The current strategy serves as an important complement to MCSF: (1) it provides strong restraints for the secondary structure of RNA ES, which is particularly useful when suitable ES-trapping mutants are not available; (2) a mutation usually causes chemical shift changes of the nearby nucleotides, and our chemical shift prediction method can account for such changes between the ES-trapping mutant and the wild-type ES. When BP-triplets involve non-GU mismatches or open base pairs, we can design additional RNA constructs containing desired BP-triplets, which is easier to implement and can be viewed as an extension of MCSF.

## Methods

**Sample preparation.** Unlabeled and [13]C/[15]N-uniformly labeled RNA samples were prepared by in vitro transcription using synthetic DNA templates (Genewiz), in-house purified T7 RNA polymerase, and unlabeled (Aladdin) or [13]C/[15]N-labeled nucleotide triphosphates (Cambridge Isotope Laboratories). The samples were purified by 15% denaturing PAGE (polyacrylamide gel electrophoresis) in 8 M urea and 1× TBE buffer, and then eluted by a "crush and soak" procedure in the corresponding buffer (20 mM Tris-HCl, 0.3 M sodium acetate, 1 mM EDTA, pH 7.4). RNAs were subsequently buffer exchanged into NMR buffer (10 mM sodium phosphate, 0.01 mM EDTA, pH 6.4) and concentrated to 250 μL using ultracentrifugal filter units with 3 KDa cutoff (Sartorius). These samples were refolded by heating at 95 °C for 5–10 min and rapidly cooled down in ice. For each sample, 1 μL of 20 mM 4,4-dimethyl-4-silapentane-1-sulfonic acid (DSS) was added as the chemical shift reference compound, and 8% $D_2O$ was added for the purpose of signal locking.

**NMR spectroscopy and data analysis.** All NMR experiments were carried out on Bruker Avance 600 MHz or 800 MHz spectrometer equipped with 5 mm triple-resonance TCI cryogenic probe. The samples were measured at 10 °C unless otherwise specified.

*Resonance assignments.* A 2D [1]H–[15]N SOFAST HMQC spectrum and a 2D [1]H–[1]H NOESY spectrum with 180 ms mixing time were recorded for imino resonance assignment for each hairpin. The secondary structure was determined unambiguously by imino–imino NOE cross-peaks, and further validated by the characteristic cross-peaks between H2 of adenines and neighboring imino protons of guanines and uridines. All spectra were processed and analyzed using NMRPipe[54] and Sparky[55].

*Analysis of NMR chemical shift data.* BMRB entries in NMR-STAR 3.1 format (Supplementary Tables 1 and 3) were processed by a set of python scripts written in-house to extract imino chemical shifts and the associated PDB access numbers. The proper PDB structures were analyzed by DSSR[39] to obtain information with regards to base pair type. A text file was then created for each BMRB entry (referred to as cs file), which relates every BP-triplet in this sample with chemical shifts of the corresponding residue in the central base pair. The hairpin RNA data acquired in this work (Supplementary Tables 1 and 3) were processed using the same pipeline, except that the base pair type information was generated by manual input based on the secondary structure rather than by DSSR. To carry out chemical shift re-referencing during the processing of training datasets, all these cs files were processed individually and the hairpin RNA data that we collected were treated first. The first cs file was re-referenced using DSS as the reference compound, and was used as the initial BP-triplet lookup table that will be filled with each BP-triplet and the associated average imino chemical shifts calculated from all occurrences of this BP-triplet. Other cs files were aligned with the BP-triplet table one by one, and the BP-triplet table was updated after each cs file was successfully aligned. The alignment was achieved by minimizing the overall chemical shift difference of common motifs (BP-triplets and UUCG apical loop) between the current BP-triplet table and each cs file. The alignment procedure of all datasets was iterated

for three rounds to guarantee convergence. During each alignment, any BP-triplet chemical shifts deviating from the mean value by more than three times the rms error were trimmed.

*[15]N $R_{1\rho}$ relaxation dispersion.* Spin-lock powers were calibrated using a modified version of $R_{1\rho}$ pulse sequence[31]. Off-resonance $R_{1\rho}$ RD profiles with different offset frequencies were recorded under spin-lock powers ($\omega_{SL}/2\pi$) ranging from 100 to 300 Hz (Supplementary Table 9). The magnetization of the N1 or N3 spin of interest undergoes relaxation for various durations ranging from 0 to 60 ms for the P5abc sample. All spectra were processed using NMRPipe and autofit script to extract intensities.

*$R_{1\rho}$ data analysis.* $R_{1\rho}$ rates under various spin-lock powers and offsets were obtained by fitting peak intensities to a mono-exponential curve. Fitting errors were estimated using Monte Carlo simulation with 50 iterations. All the on- and off-resonance $R_{1\rho}$ data were globally fitted to Laguerre equation[56].

*[1]H[N] CEST experiments.* TROSY L-optimized spin-state selective [1]H[N] CEST experiment[49] was performed with weak B1 field of 60 Hz and mixing time of 500 ms for the P5abc sample at 10 °C. A series of pseudo-3D spectra were acquired under a weak B1 field with varied offset frequencies ranging from 8.5 to 15.5 p.p.m. in step size of 30 Hz. Each 3D spectrum contains two 2D spectra corresponding to the magnetization transfer pathway of N[α] component and N[β] component, respectively.

*[1]H[N] CEST data analysis.* All NMR data were processed and analyzed using NMRPipe. The baseline of each CEST profile from N[α] or N[β] component was rescaled to 1.0 with a reference plane measured by placing B1 at far-off resonance frequency (−12 kHz). The difference profile between N[α]-derived profile and N[β]-derived profile was calculated and fitted using a python package named *ChemEx* (https://github.com/gbouvignies/chemex). The Δω value of each [1]H[N] spin was individually fitted by fixing the $k_{ex}$ and $p_b$ values, which were obtained from the globally fitting of [15]N $R_{1\rho}$ RD data for all residues involved in the concerted exchange process.

*[15]N CEST experiments and data analysis.* [15]N CEST experiment was performed with a weak B1 field of 30 Hz and mixing time of 400 ms for the P5abc sample at 10 °C. CEST profiles were recorded using an offset list ranging from 136 to 169 p.p.m. with step size 0.5 p.p.m. NMR data were analyzed the same way as described in [1]H[N] CEST data analysis.

**Calculation of ring-current shift.** To perform semiempirical calculations, RNA structures were obtained from three sources. Two sets of structures were built by RNAComposer[29] and 3DNA[57], respectively. For RNAComposer, two 312-nt RNA hairpins were generated to cover all 144 BP-triplets. For 3DNA, 64 BP-triplet fragments were generated using *fiber* command that can model only WC base pairs. The third set of structures were downloaded from PDB database using criteria of "X-ray diffraction" and resolution ≤2.0 Å, and subsequently analyzed by DSSR, resulting in 108 BP-triplets. For both crystal and 3DNA structures, Amber18 (ref. [58]) was employed to add hydrogens, as well as to perform energy minimization with the heavy atoms restrained (force constant 500 kcal mol[−1] Å[−2]).

RC Shifts were calculated by the Johnson–Bovey model[59]. The RC shielding from multiple aromatic rings of surrounding nucleotides is given by

$$\sigma_{rc} = \sum_j i_j B_j G_j(r) \qquad (1)$$

where $\sigma_{rc}$ is the RC shielding at the position in question; Σ represents the summation over contributions of aromatic rings in surrounding residues; $i_j$ is the RC intensity factor of ring $j$, representing the RC intensity ratio between ring $j$ and a reference benzene ring. $B_j$ is the shielding contribution of a single ring:

$$B_j = 3e^2/(6\pi m_e a_j c^2) \qquad (2)$$

where $e$, $m_e$, and $c$ have their conventional physical meanings; $a_j$ is the radius of ring $j$, and in our calculations the radii of 1.39 and 1.182 Å are used for six- and five-membered rings, respectively. $G_j(r)$ is the geometry factor given by

$$G(r) = \frac{1}{\sqrt{(1+\rho)^2+z_-^2}}\left[K(k_-^2) + \frac{1-\rho^2-z_-^2}{(1-\rho)^2+z_-^2}E(k_-^2)\right]$$
$$+ \frac{1}{\sqrt{(1+\rho)^2+z_+^2}}\left[K(k_+^2) + \frac{1-\rho^2-z_+^2}{(1-\rho)^2+z_+^2}E(k_+^2)\right] \qquad (3)$$

where $\rho$ and $z$ are the cylindrical coordinates with respect to the center of ring $j$, measured in the ratio relative to radius $a$; $z_{\pm} = z \pm \bar{z}$, where $\bar{z}$ is the theoretical average distance for $2p_z$ Slater orbitals from the base plane, and 0.64 Å is used here; $K(k)$ and $E(k)$ are complete elliptic integrals of the first and second kind, respectively, with modulus $k_{\pm} = \sqrt{\frac{4\rho}{(1+\rho)^2+z_{\pm}^2}}$. Finally, the RC shift can be derived

from the shielding constant in a straightforward manner:

$$\delta_{rc} = \frac{\omega_{rc} - \omega_0}{\omega_0} \times 10^6 = \frac{\omega_0(1 - \sigma_{rc}) - \omega_0}{\omega_0} \times 10^6 = -\sigma_{rc} \times 10^6 \qquad (4)$$

RC calculations were conducted initially using DC parameter set (Table 1 and Fig. 3a). GP and MV parameter sets have also been tested. GP gives rise to similar prediction r.m.s.d. values, with the $^{15}$N prediction marginally worse than the result of DC set. MV set results in a similar prediction r.m.s.d. for $^1$H$^N$, but the $^{15}$N prediction is much worse. Therefore, we chose DC set for the following calculations. The intensity factors for $^{15}$N in DC set were later calibrated (Table 1 and Fig. 3b) by using a nonlinear optimization solver that minimizes

$$\sum \left( \delta_{rc} - \delta_{BP-triplet} \right)^2 \qquad (5)$$

where $\delta_{rc}$ is the RC shift calculated using the corresponding BP-triplet structure built by RNAComposer; $\delta_{BP-triplet}$ is the chemical shift of the BP-triplet in the lookup table; $\sum$ represents the summation over all 144 BP-triplets.

**Calculation of electric-field-induced shift for proton**. EF effect arises from distant polar groups that polarize the H–X bond (X represents C or N) through the EF, thereby decreasing or increasing the local chemical shift. The chemical shift contribution from electric polarization is proportional to the local EF projected to the X–H bond, and is given by

$$\delta_{ef} = -\sigma_{ef} \times 10^6 = -A \cdot E(XH) \times 10^6 \qquad (6)$$

where $A$ is the coefficient and $-2.98 \times 10^{-12}$ esu$^{-1}$ is used here[20]; $E$ is the EF projected to H–X bond and can be calculated using Coulomb's law with the involved partial charges of polar groups taken from Amber ff94 force field[60]. The contribution of higher-order terms is considered smaller and thus negligible.

**Decomposition of BP-triplet chemical shifts**. The BP-triplet chemical shift can be decomposed according to the following formula:

$$\delta_{calc} = \delta_{intrin} + \delta_5 + \delta_3 \qquad (7)$$

where $\delta_{intrin}$, representing intrinsic chemical shift, is the contribution from the central base pair; $\delta_5$ and $\delta_3$ are the contributions from 5′ and 3′ neighboring base pairs, respectively. These three terms can be decomposed from chemical shifts of 144 BP-triplets in the lookup table, according to the procedure detailed below.

Let us take the $^{15}$N chemical shift as an example, and the processing of $^1$H$^N$ data is exactly the same. We first fixed the central base pair, as well as one of the neighboring base pairs, and obtained $^{15}$N chemical shifts of six BP-triplets from the lookup table (corresponding to varied base pairs in the other neighboring base pair). Meanwhile, the RC shifts of the varied base pairs in these six BP-triplets as probed by the central base pair were calculated separately. The two sets of $^{15}$N chemical shifts are assumed to differ by a fixed $^{15}$N offset, corresponding to contributions from the fixed neighboring base pair and the central base pair. Then this offset was subtracted from each of the six $^{15}$N chemical shifts so that their average matches the mean value of calculated RC shifts from the other neighboring base pair, resulting in six decomposed contributions of varied neighboring base pairs against the specific central base pair. The same treatment can be performed six times by altering the fixed neighboring base pair. For each specific combination of the central base pair and one of the varied neighboring base pairs, we ended up with six values and the mean value is the contribution of the neighboring base pair against the central base pair. Following this line, we obtained 24 contributions from 5′ base pairs and 24 contributions from 3′ base pairs. The intrinsic chemical shift of a given central base pair can be acquired by subtracting 5′ and 3′ contributions from corresponding experimental chemical shifts and averaging the remained shifts. For a given central base pair, 36 intrinsic chemical shifts were produced this way. The average result is shown in the table (see Table 2).

**Reporting summary**. Further information on research design is available in the Nature Research Reporting Summary linked to this article.

## Data availability
The structure coordinates used in our analyses are available at the RCSB PDB with accession codes: 1GID, 2KOC, 2M4Q, 5IEM, 1F7Y, and 5Y85. The $^1$H$^N$–$^{15}$N assignments of the hairpin RNAs in the training and testing datasets have been deposited in the BMRB under accession codes: 50018, 50029, and 50036–50073. All other data that support the findings of this study are available from the corresponding author upon reasonable request.

## Code availability
The source code of the imino chemical shift predictor and a link for the online webserver are available at https://github.com/snowrecall/csmotif-RNA. Other code used to perform calculations of this study is available upon reasonable request to the corresponding author.

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

## Acknowledgements

We thank Dr. Ning Xu in the BioNMR facility of the China National Center for Protein Sciences Beijing, for providing facility assistance. We thank Dr. Hashim Al-Hashimi for the insightful suggestions and warm help on this project. We also thank Dr. Pei Zhou and Dr. Qinglin Wu for the helpful discussion on proton CEST, and Dr. Jun Liu for comments on the manuscript. This project was supported by funds from the Tsinghua-Peking Joint Center for Life Sciences, and the Beijing Advanced Innovation Center for Structural Biology.

## Author contributions

Y.W. prepared most of samples, performed almost all NMR experiments and data analyses; G.H. and X.J. prepared several samples and collected some NMR data; T.Y. provided guidance in setting up NMR experiments; Y.X. designed and supervised the research, and performed part of data analyses. The manuscript was written by Y.X. and Y.W. with significant input from T.Y.

## Competing interests

The authors declare no competing interests.
