## [Peer Review File · Nature Communications]

REVIEWER COMMENTS

Reviewer #1 (Remarks to the Author):

The authors present a model for the prediction of imino NH chemical shifts in RNA based on ring current effects in base triplets. To extend the amount of experimental imino NH chemical shift data beyond what is available in the BMRB, the authors produced 30 RNA hairpin samples to sample the most out of 144 BP-triplet possibilities. Thereby they generate a lookup table regarding the imino chemical shift of central bases located in base pair triplets. With a total of 138 datasets, the authors demonstrate, after removing outliers, the prediction of imino ¹H and ¹⁵N chemical shifts with high accuracy. The authors show and exploit a correlation between ring-current effects of neighbouring bases and the observed imino NH chemical shifts. Finally, to demonstrate the utility of their approach they show that the predicted chemical shift prediction in combination with ¹⁵N relaxation dispersion and ¹H CEST experiments the previously suggested presence of a GG mismatch in an excited state of the Tetrahymena group I intron ribozyme P5abc can be confirmed.

The manuscript is interesting and presents a useful tool to predict imino chemical shifts in A-form RNA to aid NMR studies. Although interesting the general utility is not fully clear, as the method will mainly/only be applicable to A-form RNA, for which already a number of chemical shift-based prediction tools exist. Also, the application to study dynamics of RNA is certainly interesting but will likely be limited to few cases as most often conformational dynamics affects non-A-form helical regions, where iminos are either not observable or where the prediction will not be reliable. The authors should also provide additional support for the conclusions and provide a better validation of the method with other RNAs.

Specific comments:

- Since the data used as a training data set are derived from A-form RNA exclusively, the prediction is expected to work only for regions with A-form geometry. This creates a bias by which non-canonical conformations are neglected.
- How has the model been cross-validated, i.e. by predicting data that were not used for the training? The authors should show the results of the latest 7 BMRB data / their 10 unlabeled RNAs when using a training model derived only from their 30 RNAs vs a training model derived from the BMRB. Do the predictions agree? The authors could pick a base-pair triplet that is common to both training systems (as it is clear the BMRB is missing some base-pair triples).
- The authors use a single buffer for 30 RNA samples, and collect at a single temperature. It is known that salts, such as NaCl and MgCl₂, can affect the structure of RNA. Is it possible some bias is imposed due to the conditions they used? How did the authors account for shifting resonances as a result of temperature? It would be more reasonable to use a random array of buffers and temperatures and to include these effects in the prediction.
- RNAComposer was used to model the tertiary structures of the RNAs to better understand the correlation with ring-current. Why did the authors not use actual structures of RNAs from the PDB, as they are derived from experimental data? It is well possible that their data and predictions are biased due to the algorithms used by RNAComposer to generate the tertiary structural models. Have they considered to calculate and compare the experimental/predicted chemical shifts with quantum chemical calculations?
- Based on the graphs shown in Fig. 2, the explanation that imino NH chemical shifts are mainly influenced by ring-current effects in neighboring bases only holds for a subset of imino NH chemical shifts. Especially for ¹⁵N, the correlations seem, not unexpectedly, rather poor, indicating the presence of other effects influencing the chemical shift. This seems to affect the reliability and accuracy of the proposed method.
- The authors also mention that the prediction quality depends on which software is used to build their structural models (3DNA, vs. RNAComposer). I would strongly recommend to avoid the modelled RNAs completely and try to rely on high-resolution crystal structures, or identify the differences in the

modelling programs that affect the conformation.

- What might be the reason for the unassigned peaks in the SOFAST-HMQC spectrum of the GG1 hairpin (Fig. S3)? Could such an additional state also be present in P5abc? If so, a two-state model might not be sufficient to describe the exchange process.

- Predicted chemical shifts of BP triplets in the ES of P5abc are in good agreement with the experimentally obtained chemical shifts using additional RNA models featuring canonical BP-triplets. However, since the assumed GG mismatch is part of a non-canonical BP-triplet, those additional RNA models seem to be required to unambiguously show its presence. Hence, what is the advantage of the BP-triplet based prediction over solely using this experimental approach?

- What is the biological relevance of the presence of a GG mismatch in P5abc? It would be great if some possible roles of this are at least discussed.

- Relations between chemical shifts and excited states have been used by others (for example, for the sugar pucker, Clay, M. et al, NAR 2017) after data mining, and several nucleotides involved in excited states were evaluated.

- A variety of conformational exchange processes are possible in RNA. Since the prediction seems to require A-form geometry in both GS and ES, the question arises if the approach presented in this study would be applicable to other systems and other exchange regimes between GS and ES.

- I am not convinced that the illustrated lookup table presented in Figure 2 is as valuable as it is presented for someone trying to utilize it for determining what is going on for their GS to ES RNA system. The authors demonstrate this for just one RNA. The authors should evaluate at least two other RNAs. There are several model systems, including HIV-1 TAR RNA, where there is plenty of published work. The data should be compared to ¹³C relaxation data; is one more favorable than the other? Or can the excited state be explicitly determined using ¹⁵N vs ¹³C?

- Related to this, making mutations to validate the excited state is critical. The authors should not down-play the utility and importance of this. Instead, they could verify newly predicted examples of excited states by mutations to trap the excited state identified.

- It is possible to assign imino spectra automatically already; what statistics, if any, were used to do this and how does this compare to their data (ie RNA-PAIRS: RNA probabilistic assignment of imino resonance shifts. Bahrami, A. et al., J Biomol NMR 2012, and From raw data to protein backbone chemical shifts using NMRfx processing and NMRViewJ analysis. Johnson, B. Protein NMR. 2017).

- References to other literature linking RNA chemical shifts and conformation should be added, i.e.

 - Aeschbacher et al Nucleic Acids Res. 2013 doi: 10.1093/nar/gkt665;

 - Farès C, Amata I, Carlomagno T, J Am Chem Soc. 2007;

 - Sripakdeevong et al Nat Methods. 2014 doi: 10.1038/nmeth.2876.

- A benchmarking/comparison with existing chemical shift predictors should be done.

Minor comments:

- Can the authors comment on and discuss the drawbacks of this method? What challenges are to be expected? What is necessary for sample preparation?

- The structure of the introduction is poor. It does not adequately prepare the reader for the work they have done. The introduction should outline the background to their work. What is known about chemical shifts of RNA in general, referring to relevant literature? In the first paragraph of the introduction, the authors speak mostly of proteins, and limit descriptions for RNA chemical shift prediction to a single sentence. As they are proposing a method based on chemical shifts for RNA, it would be more appropriate to discuss, in more detail, what is known to date regarding about RNA chemical shifts.

- In the same sense, it might prove more useful to also briefly discuss what is known about ring-current contributions. And last, excited states should be discussed. The introduction should be more thorough.

- There are several grammatical errors throughout the text, making some things difficult to follow. The authors should have the manuscript edited by an outside party.

- The references are plagued with mistakes (misspellings, etc). These need to be reviewed and errors fixed.

Reviewer #2 (Remarks to the Author):

The manuscript by Wang et al. presents an excellent study in developing an NMR chemical-shift predictor for RNA imino groups and its application in studying excited states of RNA. NMR chemical shift is one of the most sensitive measurements for probing biomolecular structure and dynamics. Developing the ability to predict chemical shifts from structures, and vice versa, have been a major endeavor in the biomolecular NMR field. However, it still remains challenging to accurately predict chemical shifts in RNA. In the present study, the authors focused on developing a chemical-shift predictor for RNA imino groups, which are key in defining base pair types and secondary structures of RNA. To accomplish this, the authors designed a comprehensive set of 30 RNA constructs to cover all 144 variants of the base pair triplet (BP-triplet), the basic structural module for their prediction. Together with chemical shifts deposited in BMRB, the authors developed a database-based predictor with an excellent accuracy for both nitrogen and proton chemical shifts of the imino groups. Using their predictor, the authors unveiled that ring-current effect dominates chemical shifts of the imino groups. Furthermore, to demonstrate the utility of their predictor, the authors combined predicted chemical shifts with those extracted from NMR relaxation dispersion measurements to resolve a key structural feature of the excited state of the P5abc domain in group I intron ribozyme. Overall, the study is very well designed and executed, the manuscript is very well written, and the results are of significant interests to the readers of Nature Communications. I highly recommend publication of this excellent work.

Some minor points for the authors to consider to improve the clarity of their manuscript.

1. For GU and UG in Figure 3, while ^{15}N chemical shifts do not show good correlations between calculated and predicted values, which the authors have discussed, ^1H chemical shifts actually agree well. Can the authors comment on the different behaviors between nitrogen and proton chemical shifts?
2. The predicted chemical shifts from G•G mismatch agree well with the experimental values. However, it is not clear how these chemical shifts were predicted. Is this mismatch part of the predictor, or is the chemical shifts calculated from semi-empirical model? The authors need to clarify this.
3. On pages 5-6, the authors stated that "As shown in this map, imino resonances from GC or GU show larger dispersion while those from UG are dispersed the least.". Did the authors mean imino resonances of guanines from GC or GU and those of uridines from UG?
4. On pages 8-9, the authors might want to update the x-axis label in Figure 4a to be consistent with those in Figure 3.
5. On page 11, the authors stated that "and a hairpin with penta-loop (Supplementary Fig. 4) for U167ES where an AU mismatch is included." It seems that the AU is a canonical Watson-Crick base pair.
6. It would be beneficial to include errors for the averaged values in Table S2.

Reviewer #3 (Remarks to the Author):

The work describes a very important aspect of RNA NMR-based structural biology, namely the predictability of ^1H , ^{15}N chemical shifts of imino pairs in base-paired regions of RNA elements. The work is very important, and I recommend publication of the data. There are however some aspects that need to be taken into account.

In all figures, chemical shift regions of U and of G iminos should be plotted and correlated separately (e.g. figure 1)

Data for figure 7 is not at all so exciting since jumps from canonical to not-canonical base pairing can be derived from existing experimental chemical shift data. Thus, I would take out figure 7.

Figure 2/4: the four boxed regions should be shown separately so that one can see more. As such, the figure is unclear to read.

Response to Reviewers

We thank all the reviewers for their careful reviews and insightful comments/suggestions to our manuscript. The manuscript has been substantially revised. The changes made to the manuscript are summarized as follows.

1. We added three new subsections in Results, namely “Experimental imino chemical shifts of BP-triplets are decomposable”, “Semi-empirical method is applicable to non-canonical motifs”, and “Characterizing excited states of NRS23 RNA”. Accordingly, the newly prepared Fig. 4, 5, 8; Tab. 2; Fig. S10, S11, S15, S16, S17, S18; Tab. S6, S10, S11, and a supplementary document “rigidbody.pdf” have been added into these subsections. The Abstract has been modified to reflect these changes.
2. The original subsection “Imino chemical shift prediction helps to determine RNA excited states” has been split into two subsections, namely “Imino chemical shift prediction helps to determine RNA excited states” and “Characterizing excited states of P5abc RNA”.
3. The original Fig. 4 and Fig. 7 have been taken out of the main text. The former now becomes Fig. S9. The original Tab. 1 has been modified to include the calibrated ^{15}N ring-current parameters and moved from Methods to Results.
4. To address Reviewers’ concerns, the Discussion section has been rewritten, and the Introduction section and other main text have been modified. Fig. S2, S3, S4, S5, S6, S7, S8, and Tab. S4 have been added, and Tab. S2 has been modified. See the separate document for details.
5. A new subsection, “Decomposition of BP-triplet chemical shifts,” has been added into Methods.
6. References #11-12, 16-18, 21-22, 30, 34, 40, 48, 50-53, 59-60 have been added.

Our point-to-point responses to the reviewers’ comments are shown in bold below.

Reviewer #1:

The authors present a model for the prediction of imino NH chemical shifts in RNA based on ring current effects in base triplets. To extend the amount of experimental imino NH chemical shift data beyond what is available in the BMRB, the authors produced 30 RNA hairpin samples to sample the most out of 144 BP-triplet possibilities. Thereby they generate a lookup table regarding the imino chemical shift of central bases located in base pair triplets. With a total of 138 datasets, the authors demonstrate, after removing outliers, the prediction of imino ^1H and ^{15}N chemical shifts with high accuracy. The authors show and exploit a correlation between ring-current effects of neighbouring bases and the observed imino NH chemical shifts. Finally, to demonstrate the utility of their approach the show that the predicted chemical shift prediction in combination with ^{15}N relaxation dispersion and ^1H CEST experiments the previously suggested presence of a GG mismatch in an excited state of the Tetrahymena group I intron ribozyme P5abc can be confirmed.

The manuscript is interesting and presents a useful tool to predict imino chemical shifts in A-form RNA to aid NMR studies. Although interesting the general utility is not fully clear, as the method

will mainly/only be applicable to A-form RNA, for which already a number of chemical shift-based prediction tools exist. Also, the application to study dynamics of RNA is certainly interesting but will likely be limited to few cases as most often conformational dynamics affects non-A-form helical regions, where iminos are either not observable or where the prediction will not be reliable. The authors should also provide additional support for the conclusions and provide a better validation of the method with other RNAs.

Thanks the reviewer for all these detailed and thoughtful comments, critiques, and suggestions. It indeed makes our work stronger. Our responses are itemized as follows.

Specific comments:

1. *Since the data used as a training data set are derived from A-form RNA exclusively, the prediction is expected to work only for regions with A-form geometry. This creates a bias by which non-canonical conformations are neglected.*

We appreciate the reviewer's concern. Our work focuses on the imino chemical shifts, and most of them come from the A-form helical region. It is true that our lookup table is limited to A-form BP-triplet consisting of GC/AU/GU base pairs, and not applicable to BP-triplets involving the non-canonical base pair or adjacent to the loop region. In the original manuscript, we proposed two solutions for it. One is to make additional samples to obtain the desired non-canonical BP-triplets. This is just a workaround and is suitable for situations like characterizing ES, as it is not feasible to obtain all possible non-canonical BP-triplets by making new samples. However, when more and more non-canonical BP-triplets are measured, it is possible to develop an empirical model to predict the general cases of the non-canonical BP-triplet. The second solution, which is more important in our opinion, is the semi-empirical method. In the revised manuscript, we have explored this method further. We first calibrated ring-current (RC) parameters for ^{15}N , and then performed the semi-empirical calculation for the UUCG motif. We chose this non-canonical motif because the UUCG tetraloop is one of the most stable motifs in the RNA structure, and thus structural models with high-resolution are available. The result of UUCG motif is very encouraging (Fig. 5). Further, we applied the semi-empirical method to another non-canonical BP-triplet in P5abc^{ES}, and resulted in a reasonably good agreement with the experimental value (the black resonance in Fig. 7d). In short, an accurate prediction of RNA imino chemical shifts is quite challenging, and solving this problem once and for all is our long-term goal. I would like to emphasize here that we have made important progress in this regard.

2. *How has the model been cross-validated, i.e., by predicting data that were not used for the training? The authors should show the results of the latest 7 BMRB data / their 10 unlabeled RNAs when using a training model derived only from their 30 RNAs vs a training model derived from the BMRB. Do the predictions agree? The authors could pick a base-pair triple that is common to both training systems (as it is clear the BMRB is missing some base-pair triples).*

We cross-validated the model by evaluating the performance of the predictor against different training/testing datasets. In the new supplementary table 4, the first row (the

original training and testing datasets) and the second row (using the 30 hairpins as the training dataset and BMRB as the testing dataset) have been shown in the original manuscript. To better address the reviewer's concern, we further cross-validated two additional cases as suggested. The results are shown in the third and fourth rows of Tab. S4, in which the 30 hairpins or BMRB serves as the training dataset separately. As it turned out, no matter which strategy is used to split training/testing datasets, the achieved prediction accuracy does not change markedly. These results have been discussed in the revised manuscript (Paragraph 2 on Page 5). One may notice that the rmsd value is slightly higher when using BMRB as the training dataset (i.e. the fourth row). This is very reasonable because in BMRB dataset many BP-triplets appear only once, leading to increased uncertainty, while in the 30-hairpin dataset we have guaranteed that each BP-triplet appears at least twice.

3. *The authors use a single buffer for 30 RNA samples, and collect at a single temperature. It is known that salts, such as NaCl and MgCl₂, can affect the structure of RNA. Is it possible some bias is imposed due to the conditions they used? How did the authors account for shifting resonances as a result of temperature? It would be more reasonable to use a random array of buffers and temperatures and to include these effects in the prediction.*

First of all, we have made use of lots of BMRB data when building the training and testing datasets, and achieved satisfactory prediction accuracy. This immediately suggests the varied temperature and salt concentrations do not greatly affect the accuracy of the predictor. This conclusion is further supported by the fact that rmsd of the testing data does not change markedly when altering the strategy for splitting the training and testing datasets (see Tab. S4). What is the reason for it? It is very likely that the change in temperature or the salt concentration results in a roughly uniform trend in the shifting of each resonance, which is then 'absorbed' by our re-referencing scheme. To confirm it, we collected 2D imino spectra at 10°C and 25°C for nine hairpin samples. It turned out that all imino resonances show a sizable and largely uniform upfield shift on the ¹H dimension from 10 °C to 25 °C (Fig. S3). After re-referencing, the chemical shifts at two temperatures agree with each other very nicely: rmsd(¹⁵N) = 0.079 ppm, rmsd(¹H^N) = 0.029 ppm (i.e. well below the uncertainty of our predictor). We discussed all of these in the revised manuscript (Paragraph 2 on Page 5).

4. *RNAComposer was used to model the tertiary structures of the RNAs to better understand the correlation with ring-current. Why did the authors not use actual structures of RNAs from the PDB, as they are derived from experimental data? It is well possible that their data and predictions are biased due to the algorithms used by RNAComposer to generate the tertiary structural models. Have they considered to calculate and compare the experimental/predicted chemical shifts with quantum chemical calculations?*

We thank the reviewer for these insightful suggestions. We did not use crystal structures as models in the original manuscript, because we were unsure how accurate the A-form helical fragments in crystal structures are. But it is certainly worth a testing. In the revised manuscript, we calculated ring-current (RC) shifts using BP-triplet fragments extracted from RNA crystal structures using criteria of resolution ≤ 2.0 Å, as well as using A-form helix models built by 3DNA that can model only Watson-Crick base pairs. Compared with RNAComposer structures, the crystal structures lead to a

moderately worse agreement with the BP-triplet lookup table (Fig. S7a), whereas the result of the 3DNA model shows slightly better agreement (Fig. S7b,c). We subsequently analyzed the rigid-body parameters of these BP-triplet models (see “rigidbody.pdf” in *Supplementary Information*). We found that although the rigid-body geometries of RNAComposer model markedly deviate from those of crystal structures (especially for BP-triplets involving GU wobble), the resulting RC shifts do not differ much from that of crystal structures (Fig. S8). Moreover, it is worth noting that only 108 BP-triplets can be extracted from crystal structures. Taken together, we chose the BP-triplet models built by RNAComposer for the RC calculations. The results also underline the potential of the semi-empirical method in the structure refinement of nucleic acids. We have included these results and the relevant discussion in the main text of the revised manuscript (the last paragraph on Page 8).

With regard to quantum chemical calculations of the chemical shift, there are a number of prior works in the literature. We are also making lots of attempts in this regard. The QM calculation is undoubtedly an important tool. Its prediction accuracy, nevertheless, is remarkably lower than that of the empirical or semi-empirical methods (*JBNMR 2015*, 63:125–139; *Chem. Eur. J.* 2012, 18: 12372).

5. *Based on the graphs shown in Fig. 2, the explanation that imino NH chemical shifts are mainly influenced by ring-current effects in neighboring bases only holds for a subset of imino NH chemical shifts. Especially for ^{15}N , the correlations seem, not unexpectedly, rather poor, indicating the presence of other effects influencing the chemical shift. This seems to affect the reliability and accuracy of the proposed method.*

As we mentioned in our main text of the original manuscript, the poor prediction of ^{15}N spin of guanine and uridine in GU base pair “could be attributed to several factors such as the dynamics of GU base pair, inappropriate base plane geometry, and not fully optimized RC parameters (particularly for ^{15}N)”. We noticed that the published parameter set used for the RC calculation was parameterized against proton, and thus may not be applicable to the ^{15}N spin. In the revised manuscript, we calibrated the RC parameter set of ^{15}N using chemical shifts in the lookup table. The resulting ^{15}N parameters differ from the ^1H parameters significantly (Table 1), and are able to noticeably improve the agreement between calculated RC shifts and the lookup table (Fig. 3b). The effectiveness of the new ^{15}N RC parameters was further confirmed in the semi-empirical calculation of the UUCG motif that we are going to discuss right away (see the comparison between Fig. 5 and Fig. S10). Besides, with the semi-empirical method, we calculated chemical shifts of two non-canonical motifs (the UUCG tetraloop and a pentaloop in P5abc^{ES}) using structure models built from different software as well as X-ray and NMR structures (see Fig. 5, Tab. S6, and Fig. 7d), and obtained very encouraging results. Meanwhile, these results indicate that the semi-empirical method requires accurate 3D structures as input. Moreover, the EF contributions are not negligible in some cases (see the first paragraph on Page 17). In the end, there is clearly room for further improvement in this method: for example, EF parameters for ^{15}N (and ^{13}C), and development of reliable approaches to constructing the structural ensemble for a motif with high dynamics. The relevant discussion has been included in the main text (see *Discussion*).

6. The authors also mention that the prediction quality depends on which software is used to build their structural models (3DNA, vs. RNAComposer). I would strongly recommend to avoid the modelled RNAs completely and try to rely on high-resolution crystal structures, or identify the differences in the modelling programs that affect the conformation.

This comment has been addressed in the response to Question #4 above.

7. What might be the reason for the unassigned peaks in the SOFAST-HMQC spectrum of the GG1 hairpin (i.e. Fig. S3)? Could such an additional state also be present in P5abc? If so, a two-state model might not be sufficient to describe the exchange process.

The two unassigned resonances between 10 ppm and 11 ppm (orange arrows in the new Fig. S13) is likely due to the alternative local structure around G•G mismatch. As we have discussed in the original manuscript, the BP-triplet of this G•G mismatch in the P5abc^{ES} is not the same as that produced by the GG1 hairpin, because in the GG1 hairpin the G and U adjacent to G•G form a wobble base pair (Fig. S13a), whereas in the P5abc^{ES} the G and U are located in a three-way junction and not likely form a wobble (Fig. 7a). In line with this, we did not observe any signs of the alternative ES in the R1rho and CEST profiles of G164 and G176 (Fig. 7b, c). Therefore, our experimental results indicate the ES we observed is at least the major component of multiple potential ESs, and a two-state model is good enough to describe the exchange process.

8. Predicted chemical shifts of BP triplets in the ES of P5abc are in good agreement with the experimentally obtained chemical shifts using additional RNA models featuring canonical BP-triplets. However, since the assumed GG mismatch is part of a non-canonical BP-triplet, those additional RNA models seem to required to unambiguously show its presence. Hence, what is the advantage of the BP-triplet based prediction over solely using this experimental approach?

This is a good question. Making additional samples to obtain the chemical shifts of the desired non-canonical BP-triplet is a workaround for us, as our current BP-triplet table does not contain non-canonical motifs. In this sense, this strategy can be used without our BP-triplet lookup table, and serves as an extension of MCSF (see *Discussion* in the revised manuscript). On the other hand, the obtained chemical shifts of the non-canonical motifs can be added into the BP-triplet lookup table, and constitute an 'extended' lookup table. An accumulation of such data will eventually allow the development of empirical methods to reliably predict chemical shifts of non-canonical BP-triplets. More importantly, we proposed that the semi-empirical method can be employed to calculate the chemical shifts of non-canonical BP triplets, as long as an accurate structural model can be provided. Please note that the semi-empirical calculation can be facilitated by the decomposed chemical shift components derived from our BP-triplet table (see Tab. 2). In the revised manuscript, we have demonstrated the great potential of this method in the chemical shift calculations of the UUCG motif as well as the pentaloop motif in the P5abc^{ES} (Fig. 7). We did not test this method on the G•G mismatch simply because the corresponding structural model is absent. As a side note, the semi-empirical calculation is valuable even if a reliable structural model is not available, as it will provide strong restraints to model an accurate structure or ensemble.

9. *What is the biological relevance of the presence of a GG mismatch in P5abc? It would be great if some possible roles of this are at least discussed.*

Following the reviewer's suggestion, we have remarked in *Discussion* on the biological relevance of the presence of a GG mismatch in P5abc. "Particularly a previously speculated non-native G•G mismatch is confirmed, which helps stabilize P5abc^{ES} as a folding intermediate. [38] Relatedly, the non-native interactions have been observed in folding intermediates of protein. [52,53]"

10. *Relations between chemical shifts and excited states have been used by others (for example, for the sugar pucker, Clay, M. et al, NAR 2017) after data mining, and several nucleotides involved in excited states were evaluated.*

Thanks for bringing up this paper to our attention (we knew this work before). After reading this paper carefully, we do not think there is a direct relationship between Clay's work and our work. In the NAR paper by Clay and coworkers, the chemical shifts of C1' and C4' in the sugar ring were found to show characteristic ranges when the nucleotide is in different puckering states: C2'-endo that is dominant in the helical region, or C3'-endo that can be often seen in the non-helical region, through the analysis of the BMRB data. Further, the authors demonstrated ¹³C R1rho targeting C1' and C4' can inform us of the sugar pucker state in RNA excited states. Our work is a different story. We first established a lookup table to accurately predict imino chemical shifts in an A-form helix consisting of GC/AU/GU base pairs. Such a reliable predictor for imino chemical shifts of RNAs is currently absent. More importantly, we found that the semi-empirical model can nicely interpret the chemical shifts in the BP-triplet lookup table, and thus can be employed to calculate chemical shifts in non-canonical motifs. Finally, through analysis of the imino chemical shifts in RNA ESs (as acquired by RD experiments targeting both ¹⁵N and ¹H^N), one can derive the information of the BP-triplet where the reporter spin resides.

11. *A variety of conformational exchange processes are possible in RNA. Since the prediction seems to require A-form geometry in both GS and ES, the question arises if the approach presented in this study would be applicable to other systems and other exchange regimes between GS and ES.*

Thanks for raising this question. Our work focuses on the imino group, which can be observed in the NH 2D spectrum given that the residue is guanine or uridine in the base-paired region. It is true that our approach for characterizing secondary structure of RNA ES requires the A-form geometry in both GS and ES, or at least base-pairing in both GS and ES if the semi-empirical method is employed. However, it should be noted that the RNA imino group has been serving as an excellent indicator of the RNA secondary structure. When an RNA undergoes secondary structure rearrangement, the base-paired regions often experience a change in BP-triplets. Therefore, our approach is widely applicable to the secondary structure switch of RNAs. It must be admitted that, compared with NH group, the CH group has the advantage in detecting the unpaired region. On the other hand, however, the resonances in NH spectrum are less overlapped and are clustered clearly according to the base pair type, which makes it suitable for measuring the secondary structure switch for even larger RNAs.

The detectable exchange regimes depend on the choice of the NMR experiment. CPMG or R1rho is suitable for the relatively faster side of the intermediate exchange regime, while CEST is suitable for the slow side of the intermediate regime. In our work, we chose $^1\text{H}^{\text{N}}$ -CEST for $^1\text{H}^{\text{N}}$ RD measurement. But one can use $^1\text{H}^{\text{N}}$ -R1rho to probe the exchange process in the sub-millisecond time scale (this likely requires costly sample deuteration). It is worth mentioning that according to our practice $^1\text{H}^{\text{N}}$ -CEST works well for measuring the exchange of $\sim 2000\text{ s}^{-1}$ or even faster when a high B1 field is applied. For the slow exchange process, often two sets of resonances appear in the spectrum simultaneously. In this scenario, our approach (i.e. chemical shift prediction) still applies.

12. *I am not convinced that the illustrated lookup table presented in Figure 2 is as valuable as it is presented for someone trying to utilize it for determining what is going on for their GS to ES RNA system. The authors demonstrate this for just one RNA. The authors should evaluate at least two other RNAs. There are several model systems, including HIV-1 TAR RNA, where there is plenty of published work. The data should be compared to ^{13}C relaxation data; is one more favorable than the other? Or can the excited state be explicitly determined using ^{15}N vs ^{13}C ?*

To address the reviewer's concern, we did the following in the revised manuscript: 1) we improved the semi-empirical method by calibrating ^{15}N ring-current parameters and utilizing the decomposed chemical shift components from the BP-triplet table; 2) we demonstrated the great potential of the semi-empirical method in predicting chemical shifts of non-canonical motifs, including the UUCG motif and a pentaloop motif in P5abc^{ES}; 3) In addition to the verification of P5abc^{ES}, we applied our approach to NRS23, an RNA fragment from Rous sarcoma virus, whose ES has never been characterized before. Apart from these, we successfully applied this approach to several hairpin RNAs in another research project of our lab (manuscript is in preparation). Therefore, we are confident in our approach. With regard to ES of RNAs in the published work, although people more often used ^{13}C as the probe, there are a few ^{15}N RD data, for instance, U38-N3 in an ES of HIV-TAR. We checked the relevant paper (*PNAS* 2014, 111:9485), and found that the U38^{ES} form a U•U mismatch and is adjacent to a C•C mismatch. The lack of a reliable structural model prevents us from performing the semi-empirical calculation.

The ^{13}C RD measurement for a relatively large RNA such as P5abc is somewhat troublesome, as the site-specific labeling might be needed. In the unpublished work that I mentioned above, we measured ^{15}N R1rho, $^1\text{H}^{\text{N}}$ -CEST, and ^{13}C R1rho for several hairpin RNAs, the results are consistent with each other. In *Discussion* of the revised manuscript, we briefly compared the *pros & cons* of ^{15}N RD and ^{13}C RD. To determine the excited state accurately, it is certainly a good idea to combine both CH RD and NH RD. And importantly, a reliable chemical shift predictor for both CH and NH is a key.

13. *Related to this, making mutations to validate the excited state is critical. The authors should not down-play the utility and importance of this. Instead, they could verify newly predicted examples of excited states by mutations to trap the excited state identified.*

We appreciate the reviewer's comment. In the revised manuscript, we have modified the text in *Discussion*.

“A critical step for ES verification is to design often more than one constructs to trap ES, a strategy named Mutate and Chemical Shift Fingerprint (MCSF). The current strategy serves as an important complement to MCSF: 1) it provides strong restraints for the secondary structure of RNA ES, which is particularly useful when suitable ES-trapping mutants are not available; 2) mutation usually causes chemical shift changes of the nearby nucleotide, and our chemical shift prediction method can validate such changes between ES-trapping mutant and wild-type ES. When BP-triplets involve non-GU mismatches or opened base pairs, we can design additional RNA constructs containing desired BP-triplets, which is easier to implement and can be viewed as an extension of MCSF.”

14. *It is possible to assign imino spectra automatically already; what statistics, if any, were used to do this and how does this compare to their data (ie RNA-PAIRS: RNA probabilistic assignment of imino resonance shifts. Bahrami, A. et al., J Biomol NMR 2012, and from raw data to protein backbone chemical shifts using NMRFX processing and NMRViewJ analysis. Johnson, B. Protein NMR. 2017).*

We noticed RNA-PAIRS long time ago, and tested it on two RNAs (P5abc and Gln-riboswitch, which are also in the testing dataset). The results turned out to be, no more than one third of the residues can be correctly assigned by RNA-PAIRS. Therefore, RNA-PAIRS apparently has room for improvement when doing assignments for relatively complicated RNAs. In contrast, combining imino NOE connectivities with our predictor would be particularly useful. According to our practice, all imino resonances of both RNAs can be assigned correctly this way.

15. *References to other literature linking RNA chemical shifts and conformation should be added, i.e. Aeschbacher et al Nucleic Acids Res. 2013 doi: 10.1093/nar/gkt665;*

Farès C, Amata I, Carlomagno T, J Am Chem Soc. 2007;

Sripakdeevong et al Nat Methods. 2014 doi: 10.1038/nmeth.2876.

These references have been added as suggested.

16. *A benchmarking/comparison with existing chemical shift predictors should be done.*

There is currently no chemical shift predictor for the RNA imino group, except for LARMOR^D that provides a tentative and experimental function for it. In the revised manuscript, we compared the predicted results between our method and LARMOR^D (Fig. S2). Not surprisingly, the performance of LARMOR^D is much worse.

Minor comments:

17. *Can the authors comment on and discuss the drawbacks of this method? What challenges are to be expected? What is necessary for sample preparation?*

We appreciated the reviewer’s comments. We have added a paragraph in *Discussion* to discuss the drawbacks of our method as well as the expected challenges.

“The imino chemical shift prediction based on our BP-triplet lookup table is only applicable to the A-form region made of WC base pair and GU wobble. For non-canonical motifs, the central base pair is typically adjacent to non-canonical base pairs, bulges, loops, and junctions. The semi-empirical method has been proven to be particularly helpful in this scenario. We found that both the RC and EF shifts are sensitive to minor conformational changes of an RNA, and thus an accurate structural description of non-canonical motif in static form or (more often) ensemble form is required for reliable chemical shift prediction. Conversely, chemical shifts can serve as highly effective structural restraints with the aid of semi-empirical method. To this end, accurate RC and EF models for ^{15}N and ^{13}C are in urgent need.”

Regarding “sample preparation”, we assume that the reviewer refers to the sample preparation for the ^{15}N R1rho and ^1H CEST experiments. The RNA samples need to be ^{15}N -uniformly (or $^{13}\text{C}/^{15}\text{N}$ -uniformly) labeled, which has been described in *Methods*.

18. *The structure of the introduction is poor. It does not adequately prepare the reader for the work they have done. The introduction should outline the background to their work. What is known about chemical shifts of RNA in general, referring to relevant literature? In the first paragraph of the introduction, the authors speak mostly of proteins, and limit descriptions for RNA chemical shift prediction to a single sentence. As they are proposing a method based on chemical shifts for RNA, it would be more appropriate to discuss, in more detail, what is known to date regarding about RNA chemical shifts.*

We have modified the *Introduction* to outline what has been done generally for chemical shift prediction of RNAs. The relevant references have been included as well. With regard to imino chemical shifts, we stated that “As a result, there are currently no predictors for the imino group of RNAs, except for a tentative functional module in program LARMOR^D.” (see Paragraph 2 on Page 3)

19. *In the same sense, it might prove more useful to also briefly discuss what is known about ring-current contributions. And last, excited states should be discussed. The introduction should be more thorough.*

In response to the reviewer’s comments, we have briefly summarized what has been done about the ring-current effect in the revised manuscript (see Paragraph 1 on Page 8). We have also edited the “*Imino chemical shift prediction helps to determine RNA excited states*” subsection and the *Discussion* section to include more introduction and discussion as well as relevant references about excited states.

20. *There are several grammatical errors throughout the text, making some things difficult to follow. The authors should have the manuscript edited by an outside party.*

We have made lots of efforts to improve the text. In addition, we have sent the manuscript to outside party to get the text polished, as suggested by the reviewer.

21. *The references are plagued with mistakes (misspellings, etc). These need to be reviewed and errors fixed.*

We thank the reviewer for catching this oversight. We have checked the references carefully and did our best to fix errors in the references.

Reviewer #2:

The manuscript by Wang et al. presents an excellent study in developing an NMR chemical-shift predictor for RNA imino groups and its application in studying excited states of RNA. NMR chemical shift is one of the most sensitive measurements for probing biomolecular structure and dynamics. Developing the ability to predict chemical shifts from structures, and vice versa, have been a major endeavor in the biomolecular NMR field. However, it still remains challenging to accurately predict chemical shifts in RNA. In the present study, the authors focused on developing a chemical-shift predictor for RNA imino groups, which are key in defining base pair types and secondary structures of RNA. To accomplish this, the authors designed a comprehensive set of 30 RNA constructs to cover all 144 variants of the base pair triplet (BP-triplet), the basic structural module for their prediction. Together with chemical shifts deposited in BMRB, the authors developed a database-based predictor with an excellent accuracy for both nitrogen and proton chemical shifts of the imino groups. Using their predictor, the authors unveiled that ring-current effect dominates chemical shifts of the imino groups. Furthermore, to demonstrate the utility of their predictor, the authors combined predicted chemical shifts with those extracted from NMR relaxation dispersion measurements to resolve a key structural feature of the excited state of the P5abc domain in group I intron ribozyme. Overall, the study is very well designed and executed, the manuscript is very well written, and the results are of significant interests to the readers of Nature Communications. I highly recommend publication of this excellent work.

Some minor points for the authors to consider to improve the clarity of their manuscript.

1. *For GU and UG in Figure 3, while ^{15}N chemical shifts do not show good correlations between calculated and predicted values, which the authors have discussed, ^1H chemical shifts actually agree well. Can the authors comment on the different behaviors between nitrogen and proton chemical shifts?*

We appreciate the reviewer's comment. We think the different behaviors between nitrogen and proton chemical shift prediction are partially due to the inappropriate ring-current (RC) parameters for nitrogen. The RC parameter set used for the calculation in Fig. 3a was parameterized against proton, and thus may not be applicable to ^{15}N spin. In the revised manuscript, we recalibrated the RC parameters of ^{15}N . The new parameters indeed improve the correlations for ^{15}N (Fig. 3b). Apart from that, other factors may play roles as well, such as dynamics of GU base pairs,

improper base plane geometry, ignorance of EF contributions. All of these have been discussed from Paragraph 2 on Page 8 through Paragraph 2 on Page 9. These factors may affect ^{15}N and $^1\text{H}^{\text{N}}$ to varying degrees. BTW, some helpful details can be found in our response to Question #5 of Reviewer 1.

2. *The predicted chemical shifts from G·G mismatch agree well with the experimental values. However, it is not clear how these chemical shifts were predicted. Is this mismatch part of the predictor, or is the chemical shifts calculated from semi-empirical model? The authors need to clarify this.*

The predicted imino chemical shifts of G176^{ES} and U164^{ES} come from a GG1 hairpin we prepared. This hairpin RNA contains the desired BP-triplets of G176^{ES} and U164^{ES}, thereby allowing the prediction of G·G mismatch. In our original Fig. 7a, we did not distinguish these “non-canonical” BP-triplets from those in the BP-triplet lookup table. This might cause the confusion. In the revised figure (i.e., Fig. 7d), we use solid blue resonances and blue resonances to distinguish them. The corresponding descriptions in the main text can be found in both the original (the last paragraph on Page 11 and the first paragraph on Page 13) and the revised manuscript (Paragraph 2 on Page 16 and the last paragraph of *Discussion*). It is worth mentioning that we made predictions for U167^{ES} using three different methods in the revised manuscript (Fig. 7d): the lookup table; the “non-canonical” BP-triplet from a hairpin sample we prepared on purpose; the semi-empirical model (a crystal structure model for U167^{ES} is available).

3. *On pages 5-6, the authors stated that “As shown in this map, imino resonances from GC or GU show larger dispersion while those from UG are dispersed the least.” Did the authors mean imino resonances of guanines from GC or GU and those of uridines from UG?*

We have modified the text following the suggestion: “*imino resonances of guanines from GC or GU*” and “*those of uridines from UA or UG*” (Paragraph 1 on Page 7).

4. *On pages 8-9, the authors might want to update the x-axis label in Figure 4a to be consistent with those in Figure 3.*

Thank the reviewer for pointing out this issue. We have updated the x-axis label of Figure 4a (i.e., Fig. S9a in the revised manuscript), and now it is consistent with other figures.

5. *On page 11, the authors stated that “and a hairpin with penta-loop (Supplementary Fig. 4) for U167ES where an AU mismatch is included.” It seems that the AU is a canonical Watson-Crick base pair.*

We are sorry for the confusion. We have changed “AU mismatch” into “A·U mismatch” to indicate that A and U do not form a canonical Watson-Crick base pair (Paragraph 2 on Page 16 and the figure caption of Fig. S14).

6. It would be beneficial to include errors for the averaged values in Table S2.

We thank the reviewer for pointing it out. We have included errors for the averaged values in Table S2 (column 3 and column 5). In addition, we have added error bars into the “chemical shift map” (Fig. 2 and Fig. S4).

Reviewer #3:

The work describes a very important aspect of RNA NMR-based structural biology, namely the predictability of ^1H , ^{15}N chemical shifts of imino pairs in based paired regions of RNA elements.

The work is very important, and I recommend publication of the data.

There are however some aspects that need to be taken into account.

1. *In all figures, chemical shift regions of U and of G iminos should be plotted and correlated separately (e.g. figure 1)*

In the revised manuscript, we have plotted the correlation of guanines and uridines separately (Fig. 1c,d and Fig. 4).

2. *Data for figure 7 is not at all so exciting since jumps from canonical to not-canonical base pairing can be derived from existing experimental chemical shift data. Thus, I would take out figure 7.*

We agree with it. The original figure 7 has been taken out of the revised manuscript.

3. *Figure 2/4: the four boxed regions should be shown separately so that one can see more. As such, the figure is unclear to read.*

Thanks a lot for the suggestion. We have plotted different regions of the "chemical shift map" in multiple enlarged figures to make it clearer to read (Fig. S4 and Fig. S9c).

REVIEWER COMMENTS

Reviewer #1 (Remarks to the Author):

The authors have addressed some of the concerns raised, but for several others, they did not really respond other than arguing that high-resolution experimental structures may be not accurate. The authors add new data, which, however, raise additional concerns. Thus, I still have some critical remaining comments, which should be addressed.

Having said and assuming that the authors address the concerns I support publication of the manuscript in Nature Communications as it provides new interesting ideas that will further advance the field.

Specific points

1. Previous comment 4: The question was why the authors have used a computational algorithm, RNAComposer, to generate models of structures rather than use structures that exist in the PDB. The authors argue that they obtain worse results when looking at experimentally-resolved structures. For a helical domain it is fair to assume that the geometry seen in a high-resolution crystal structure is a fair presentation of the lowest energy state which will dominate the chemical shift and ring current effects. I can see that this will be different for a single stranded / flexible region. It does not seem appropriate to calibrate and reference their predictor against modelled structures and ignoring high-resolution experimental data.

2. It was suggested that the authors should validate their approach on an RNA that already has been characterized, in terms of excited states. Instead of choosing something already known to show proof of principle, the authors now 1) argue that the structures are not reliable so they do not use them, and then 2) introduce a brand new RNA, NRS23 which they probe for excited states. There is NO functional data suggesting or supporting this excited state is biologically relevant. Even the model proposed is questionable (they say the complementarity is favorable, but do not use anything, not even INTA RNA to evaluate the interaction (see S18)). So it is not clear to what extent this new example supports the predictor and its utility. Rather than "squeezing" these new data into the current manuscript they should rather report this together with functional analysis separately. Instead, the authors should take data from published work, which does not require any new experiments, to demonstrate their approach in characterizing excited states.

3. The authors tested RNA-PAIRS on two RNAs, when they have a set of 30 from their lab, and find that from two data points, it is not accurate relative to what they propose. This does not seem fair, for a proper benchmark all RNAs should be compared.

4. Figure 8b for G2; I don't think that fitting is trustworthy. Same for Figure S16 G8

Reviewer #2 (Remarks to the Author):

The revised manuscript by Wang et al. has addressed my concerns of the original manuscript. Hence, I highly recommend publication of this excellent work, which should be of significant interests to the readers of Nature Communications.

Reviewer #3 (Remarks to the Author):

The authors have addressed my previous concerns. The manuscript should be accepted.

Response to Reviewers

Reviewer #1:

The authors have addressed some of the concerns raised, but for several others, they did not really respond other than arguing that high-resolution experimental structures may be not accurate. The authors add new data, which, however, raise additional concerns. Thus, I still have some critical remaining comments, which should be addressed.

Having said and assuming that the authors address the concerns I support publication of the manuscript in Nature Communications as it provides new interesting ideas that will further advance the field.

We thank the reviewer for his additional comments/suggestions to our manuscript. The point-to-point responses to the reviewers' comments are shown in bold below. And the manuscript has been revised accordingly.

Specific points

1. Previous comment 4: The question was why the authors have used a computational algorithm, RNAComposer, to generate models of structures rather than use structures that exist in the PDB. The authors argue that they obtain worse results when looking at experimentally-resolved structures. For a helical domain it is fair to assume that the geometry seen in a high-resolution crystal structure is a fair presentation of the lowest energy state which will dominate the chemical shift and ring current effects. I can see that this will be different for a single stranded / flexible region. It does not seem appropriate to calibrate and reference their predictor against modelled structures and ignoring high-resolution experimental data.

We appreciate the reviewer's comment. At first glance, it is very reasonable to assume that the A-form helical geometry parameters from the crystal structures are the most trustable ones. However, judging from the rigid-body parameters extracted from the crystal structures with resolution ≤ 2.0 Å, the A-form geometries are quite diverse even for the same BP-triplet (see "rigidbody.pdf" in *supplementary information*). As a result, the ring-current shifts calculated from the crystal BP-triplet fragments also show high diversity (see Fig. S7a in the currently revised manuscript). This diversity might arise from variations in the crystallization conditions, buffer conditions, crystal packing effects, and structure refinement algorithms. The geometry might also be affected by the existence of other components in the tertiary structure. This immediately raises a question: which crystal BP-triplet fragment should be used? In our previous revision of the manuscript, we randomly picked up a set of BP-triplets from crystal fragments, and the correlation is moderately worse than the case of RNAComposer (see Fig S7a in the previous revision). Inspired by the reviewer's comments, we speculate that taking the average of chemical shifts predicted by the same BP-triplet could lead to a better result. Indeed, the agreement with the BP-triplet lookup table gets improved to some extent (see Fig. S7b in the currently revised manuscript). Now the correlation of ^{15}N chemical shift is comparable to that of RNAComposer, but the correlation of ^1H chemical shift is still noticeably worse. Please note that only 108 BP-triplets can be

extracted from the crystal structures. To make a fair comparison, we also used the same set of 108 BP-triplets built by RNAComposer, and the conclusion is the same.

Now we have to make a decision: whether should we use BP-triplet fragments from the crystal structures or use BP-triplet fragments from RNAComposer? We tend to think RNAComposer is a better choice, for the following reasons: 1) There are only 108 BP-triplet fragments (i.e. 36 BP-triplets are missing) that can be extracted from the crystal structures. 2) RNAComposer builds 3D structures using structure elements from PDB database comprising crystal and NMR structures, and thus the geometry information of the crystal structures has been encoded into the RNAComposer models to some extent. Although we cannot say that models from RNAComposer are definitely closer to reality, we also have no evidence that the ensemble form of BP-triplets from the crystal structures can reflect the real geometry of A-form helices in solution better than RNAComposer model does. 3) The “idealized” A-form helices generated by software tools have been used in RDC fitting for the canonical A-form helical structures (for example, *JACS* 2000, 122:11561, *JMB* 2002, 315:95). According to our practice (and also the experiences in Al-Hashimi lab), this approach will result in an agreement with the experimental RDCs that is no worse than the case using crystal structures, if not better.

2. It was suggested that the authors should validate their approach on an RNA that already has been characterized, in terms of excited states. Instead of choosing something already known to show proof of principle, the authors now 1) argue that the structures are not reliable so they do not use them, and then 2) introduce a brand new RNA, NRS23 which they probe for excited states. There is NO functional data suggesting or supporting this excited state is biologically relevant. Even the model proposed is questionable (they say the complementarity is favorable, but do not use anything, not even INTA RNA to evaluate the interaction (see S18). So it is not clear to what extent this new example supports the predictor and its utility. Rather than “squeezing” these new data into the current manuscript they should rather report this together with functional analysis separately. Instead, the authors should take data from published work, which does not require any new experiments, to demonstrate their approach in characterizing excited states.

We are sorry for not making it clearer when answering the reviewer’s question in the previous revision. We understand that the reviewer wants us to demonstrate our approach using the published data of RNA excited states instead of using new samples. In fact, we did try our best to address the reviewer’s concern. After searching the literature carefully, we found that most of the prior works focus on the RD measurements targeting carbons, whereas only few works target the imino group. There is also a recent review paper summarizing the existing studies of the RNA excited states (*ChemBioChem* 2019, 20:2685). Apart from two publications about transient tautomeric and anionic WC-like GU base pair in which ¹⁵N data were measured, there are only four publications involving imino RD measurement of RNAs. (a) HIV-1 TAR (*PNAS* 2014, 111:9485); (b) P5abc (*Nature communications* 2016, 7: 1); (c) miR-34a (*Nature* 2020, 583:139); (d) Fluoride riboswitch (*Journal of Magnetic Resonance* 2020, 310: 106642). Among them, P5abc has been served as an example to demonstrate our approach. The other three RNAs, unfortunately, cannot be predicted

in terms of chemical shifts of the excited states because non-canonical BP-triples are involved. For example, U38 in TAR excited state forms a U•U mismatch and is also adjacent to a C•C mismatch. The situations of the two other RNAs are similar. According to our framework, the imino chemical shifts of these non-canonical motifs can be predicted by the semi-empirical method (BTW, making an additional sample containing the specific non-canonical motif is another solution, but it is apparent not what the reviewer wants us to do; it is also not necessary as the prior studies have made mutants to trap the excited state). This method requires a reliable 3D structure as the input as the semi-empirical calculation is sensitive to even minor changes of the conformation. However, such structural models do not exist for the excited states of these three RNAs. Therefore, the verification using the existing studies of RNA excited states is NOT possible. Given this situation, we measured a new RNA (NRS23) to fulfill the reviewer's requirement. Even leaving the data of NRS23 alone, we think that the verification of our approach is already sufficient. To show the proof of principle, it is not imperative to use the excited state data to verify the new approach. As long as it works for the ground state, there is no reason why it does not work for the excited state. We have successfully demonstrated the new approach in six variations of the UUCG motif for which reliable 3D structures are available. In addition, we applied our approach to a non-canonical BP-triplet motif of P5abc in the previous revision.

With regard to the story of NRS23 excited state, we completely agree with the reviewer's comment: we'd better report it together with the future functional analysis. To this end, we have taken the NRS23 part out of the manuscript.

3. The authors tested RNA-PAIRS on two RNAs, when they have a set of 30 from their lab, and find that from two data points, it is not accurate relative to what they propose. This does not seem fair, for a proper benchmark all RNAs should be compared.

Following the reviewer's suggestion, we assessed the performance of RNA-PAIRS on all 30 RNAs, as listed below. Judging from our result, RNA-PAIRS failed to generate automatic imino assignment to a satisfactory extent. This outcome is not that surprising, because the benchmark in the original RNA-PAIRS paper took inputs from BMRB entries instead of the real NOESY spectra. Our imino prediction tool should be helpful in developing a new generation of automatic assignment tool of RNA iminos. In fact, we are currently collaborating with NMRFAM-SPARKY team to pursue this goal.

RNA sample	% Correct best choice	% Correct assignment in top 3
HP15	61	67
HP16	12	29
HP17	78	89
HP18	59	65
HP19	56	56
HP20	47	47
HP21	44	56
HP22	94	94
HP23-GC	94	94

HP23-GU3	40	50
HP24	65	76
HP25	42	42
HP26	72	72
HP27	94	94
HP28	50	78
Rest01	94	100
Rest02	94	100
Rest03	100	100
Rest04	94	94
Rest05	88	100
Rest06	67	67
Rest07	100	100
Rest08	75	75
Rest09	94	94
Rest10	47	53
HP-GU	83	94
PL01	69	85
PL02	92	92
PL03	58	58
PL04	42	50
P5abc	6	12
Gln-riboswitch	13	33

4. *Figure 8b for G2; I don't think that fitting is trustworthy. Same for Figure S16 G8*

We assume the reviewer's concern is that the R1rho profiles of G2 and G8 are noisy. Such observations are not uncommon in RD measurement. This is mainly due to the small $\Delta\omega$ between GS and ES. It could also be partially due to the more complicated motions of these two residues during conformational exchange, as they are located in or near a loop/terminal region.

As stated in our response to question #2, all the contents of NRS23 (including Fig 8b and Fig. S16) have been removed from the manuscript.

REVIEWERS' COMMENTS

Reviewer #1 (Remarks to the Author):

The authors have addressed the remaining concerns in the rebuttal and further revisions of the manuscript. I fully support publication.

There is still some concern regarding the response by the authors on whether experimentally derived structures (NMR / Xray) are better than computationally generated structures. I remain unconvinced by this. That conformational dynamics and breathing will affect the prediction is certainly agreeable, and the fact that averaging various crystallographic structures improves the predictions is supportive of this.

Nevertheless, the approach is interesting and its utility will hopefully be seen in the future by its recognition in the field.